# Heterogeneous T cell motility behaviors emerge from a coupling between speed and turning in vivo

Elizabeth R Jerison[1]*, Stephen R Quake[1,2,3]*

[1]Department of Applied Physics, Stanford University, Stanford, United States; [2]Department of Bioengineering, Stanford University, Stanford, United States; [3]Chan Zuckerberg Biohub, San Francisco, United States

**Abstract** T cells in vivo migrate primarily via undirected random walks, but it remains unresolved how these random walks generate an efficient search. Here, we use light sheet microscopy of T cells in the larval zebrafish as a model system to study motility across large populations of cells over hours in their native context. We show that cells do not perform Levy flight; rather, there is substantial cell-to-cell variability in speed, which persists over timespans of a few hours. This variability is amplified by a correlation between speed and directional persistence, generating a characteristic cell behavioral manifold that is preserved under a perturbation to cell speeds, and seen in Mouse T cells and *Dictyostelium*. Together, these effects generate a broad range of length scales over which cells explore in vivo.

## Introduction

Many immune cells migrate through tissue in search of antigen or pathogens. In some cases, such as during extravasation from blood vessels and homing to target organs, this migration is guided by chemokine gradients (*Witt et al., 2005*; *Okada et al., 2005*; *Germain et al., 2012*; *Sarris and Sixt, 2015*). However, for naive T cells within T cell zones, in situ imaging studies have found that unguided random walk processes dominate (*Miller et al., 2002*; *Miller et al., 2003*; *Preston et al., 2006*; *Cahalan and Parker, 2008*; *Beltman et al., 2007*; *Banigan et al., 2015*; *Harris et al., 2012*; *Worbs et al., 2007*; *Textor et al., 2011*; *Beauchemin et al., 2007*; *Mrass et al., 2006*; *Katakai et al., 2013*; *Mrass et al., 2017*, reviewed in *Mrass et al., 2010*; *Krummel et al., 2016*). This observation creates a conceptual challenge: T cells must dwell at scales of microns to make contact with antigen presenting cells (*Wülfing et al., 1997*; *Krummel et al., 2000*; *Beltman et al., 2009a*; *Fricke et al., 2016*), yet migrate over scales of millimeters to find rare targets. A conventional diffusive random walk struggles to access these varied scales efficiently, since a walker that dwells near another cell for 1 min would require several days to travel 1 mm. Several authors have suggested that T cells may have an intrinsic behavioral program that allows them to explore over different length scales (*Harris et al., 2012*; *Krummel et al., 2016*; *Mempel et al., 2004*). However, testing this hypothesis via in situ fluorescence microscopy raises inherent technical challenges: to observe a single cell accessing a broad range of spatial scales, it is necessary to have micron scale resolution over fields of view of millimeters, with low enough photodamage to observe the same cells at high spatiotemporal resolution over long periods. For example, one intriguing proposal is that T cells perform Levy flight (*Harris et al., 2012*), an anomalous random walk characterized by a power-law distribution of step sizes. Such random walks have been described in detail in the physics and ecology literature (*Shlesinger et al., 1995*; *Bartumeus et al., 2005*; *Viswanathan et al., 2011*), and their scale-free behavior provides a natural way for foragers to accelerate searches in many contexts (*Bartumeus et al., 2002*). However, observation over short periods cannot distinguish between

*For correspondence:
ejerison@stanford.edu (ERJ);
steve@quake-lab.org (SRQ)

Competing interests: The authors declare that no competing interests exist.

Levy flight and heterogeneity amongst individual walkers (*Petrovskii et al., 2011*), both of which can create a broad distribution of displacements. More generally, we would like to understand whether there is a statistically-consistent behavioral program carried out by these cells.

To address this question, we used selective plane illumination microscopy (*Pitrone et al., 2013*; *Power and Huisken, 2017*) to observe the native population of T cells in the live larval zebrafish (Tg (*lck*:GFP, *nacre*$^{-/-}$) *Langenau et al., 2004*), over millimeter fields of view and periods of a few hours. We observed a population of motile cells in tissue in the tail of the zebrafish, primarily in the tail fin and larval fin fold (*Figure 1A*, *Figure 1—video 1*). We chose this population for further study because of the potential to measure the interstitial exploration behavior of the cells over long length-scales, and to dissect the variation in behavior over a populations of cells.

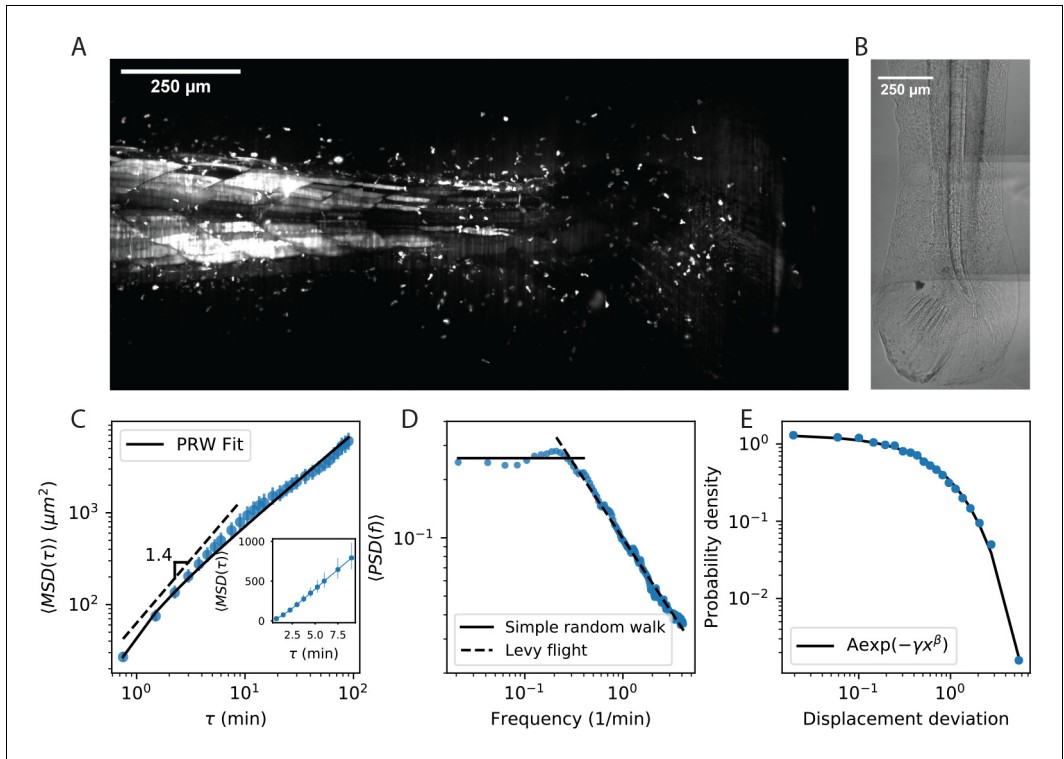

**Figure 1.** Cell motility behavior is inconsistent with Levy flight. (**A**) Maximum Z projection of a Tg(*lck*:GFP, *nacre*$^{-/-}$) zebrafish at 12 dpf. This projection represents the first frame of a timecourse; see *Figure 1—video 1*. (**B**) Brightfield of the region of tissue shown in A. Stitching across three tiles was performed in ImageJ. (**C**) Mean squared displacement as a function of time lag. The cells migrate super-diffusively on scales of a few minutes. The MSD for a persistent random walk is fit to the data (Materials and methods, Appendix 1). Error bars represent 95% confidence intervals on a bootstrap over n = 316 trajectories containing all measured time intervals. (See also *Figure 1—figure supplement 2*). Inset: linear scale for the first 10 min. (**D**) The velocity power spectrum, averaged across all trajectories (n = 634). A Levy (scale-free) process consistent with the short time behavior would result in a continuation of the high frequency slope (dashed line). Instead, we observe a timescale at a few minutes. (**E**) Distribution of bout lengths within a trajectory (Materials and methods), fit with a stretched exponential (n = 35819 bouts). The fitted stretch parameter $\beta = .9$. For all panels, trajectories were pooled from n = 16 fish.

The online version of this article includes the following video, source data, and figure supplement(s) for figure 1:

**Source data 1.** Source data for *Figure 1C*.
**Source data 2.** Source data for *Figure 1D*.
**Source data 3.** Source data for *Figure 1E*.
**Figure supplement 1.** As in *Figure 1B*, with annotations.
**Figure supplement 2.** MSD for all trajectories tracked through 15 min.
**Figure 1—video 1.** T cell dynamics in the larval zebrafish tail and fin fold.
https://elifesciences.org/articles/53933#fig1video1

Rather than a single broad distribution of speeds sampled by all cells, as in Levy flight, we observed considerable heterogeneity in both speed and turning behavior across cells. This observation, together with prior literature (*Maiuri et al., 2015*), prompted us to analyze the distribution of cell behaviors in a space defined by speed and turning statistics. Surprisingly, cell behaviors fell on a one dimensional manifold in this space, characterized by a coupling between speed and directional persistence. Analysis of previously-published data in mouse T cells (*Gérard et al., 2014*) and *Dictyostelium* (*Dang et al., 2013*) within this framework showed that their migration statistics fell along a similar manifold. Our results show that a wide variation in speeds, combined with a coupling between speed and persistence, generate a broad distribution of length scales of exploration in vivo.

## Results

### Cell motility behavior is inconsistent with Levy flight

To investigate the statistical properties of T cell motility in our system, we measured cell trajectories within the tissue posterior to the anus (Materials and methods, *Figure 1—video 1*, *Figure 2—video 1*). This region is composed primarily of the tail fin and larval fin fold, which represent a millimeter-scale tissue over which the cells can potentially migrate. We note that cells in circulation, while present, move orders of magnitude faster than those in tissue, and thus are not included in our observations or analysis. Note also that our observations were performed in the absence of an external perturbation such as an infection.

We first evaluated evidence for Levy flight behavior, as opposed to persistent random walks (*Beauchemin et al., 2007*; *Beltman et al., 2007*; *Banigan et al., 2015*; *Harris et al., 2012*), in our system. The distinction hinges on whether the statistics of individual trajectories are scale-free, so that super-diffusive behavior continues to long times; or if, alternatively, individual trajectories are diffusive at long times but there is heterogeneity across the population. To address this question, we performed a standard analysis of mean squared displacement as a function of time interval. Consistent with previous measurements (*Beauchemin et al., 2007*; *Beltman et al., 2007*; *Banigan et al., 2015*; *Harris et al., 2012*), we observed a faster-than-linear increase in MSD at early times, indicating super-diffusive behavior, with a best-fit line in surprisingly good quantitative agreement with previous observations up through 10 min (*Harris et al., 2012*; *Fricke et al., 2016*; *Figure 1C*). However, we observed a transition at the scale of minutes, consistent with persistent random walks, and inconsistent with Levy flight (also note the straight line on a linear scale, *Figure 1C* inset, characteristic of diffusive behavior). Note that while we have examined the subset of longer trajectories to measure the behavior through an additional order of magnitude in time, this result also holds when examining all trajectories through 15 min (*Figure 1—figure supplement 2*). To further test for an intermediate timescale, we computed the velocity-velocity power spectrum, using secant-approximated velocities along each trajectory (Materials and methods). This quantity captures the timescale at which the velocities become decorrelated, if it exists; for a Levy-flight process the same negative slope is observed at all frequencies (*Viswanathan et al., 2005*), while a persistent random walk model passes towards zero slope at low frequencies (*Viswanathan et al., 2005*; *Pedersen et al., 2016*). Consistent with the MSD analysis, we observe two regimes, with a clear timescale on the order of minutes (*Figure 1D*). Finally, we computed the distribution of lengths between direction changes (bout lengths) within a trajectory (Materials and methods), scaled by the average bout length as suggested in *Petrovskii et al., 2011*, and did not observe the characteristic Levy-flight power law (*Figure 1E*).

### Motility behavior is heterogeneous across cells

Since we did not find support for Levy flight in our system, we next evaluated evidence for cell-to-cell heterogeneity. From examples of velocity traces (*Figure 2A,C–E*, *Figure 2—video 1*), we observed substantial variation in speed between cells, that can persist over spans of a few hours. These trajectories are not atypical: overall, 88% of trajectories have distributions of secant-approximated speeds that are inconsistent with the speed distribution pooled on all trajectories (KS test, $p<.01$). Interestingly, we also found significant heterogeneity in cell turning behavior: 67% of cells had turn angle distributions inconsistent with the overall distribution (KS test, $p<.01$).

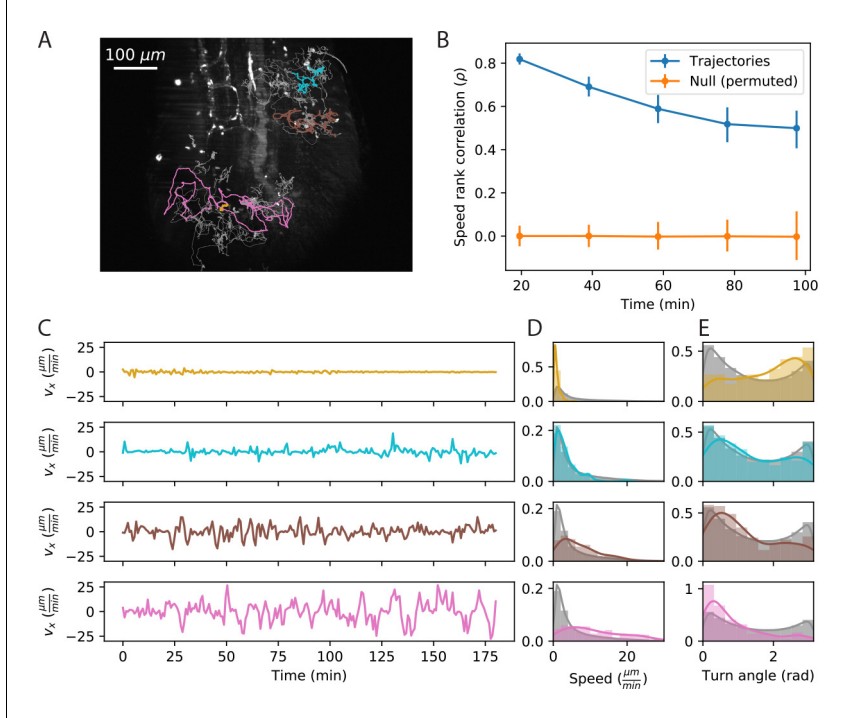

**Figure 2.** Cell speed and turning behavior are heterogeneous. (**A**) Example of trajectories recorded over 3 hr at a 12 s interval (Tg(*lck*:GFP, *nacre*⁻ᐟ⁻) zebrafish; 10 dpf). Here we show a maximum Z projection of the 900th frame with trajectories overlaid; see **Figure 2—video 1** for the timecourse. Examples of four cell trajectories, with a range of characteristic speeds, are colored. (**B**) Spearman rank correlation between trajectory speeds measured on non-overlapping 19.5 min intervals, as a function of the time between the beginning of the intervals. Error bars represent 95% confidence intervals on a bootstrap over trajectories. The null model was constructed by permuting measured speeds across all the trajectories at each interval; error bars represent 95% confidence intervals over the permutations. (Calculations performed on the n = 321 trajectories of at least 117 min in length.) Trajectories were pooled over n = 16 fish. (**C**) Velocity traces for the four cells highlighted in A. (**D**) Secant-approximated speed distributions for each cell from A, compared with the distribution over all cells (grey;n = 98141 steps). (**E**) Turn angle distributions for each cell from A, compared with the distribution over all cells (grey;n = 96122 turn angles). The online version of this article includes the following video, source data, and figure supplement(s) for figure 2:

**Source data 1.** Source data for *Figure 2B*.
**Source data 2.** Source data for *Figure 2C*.
**Source data 3.** Source data for *Figure 2D*.
**Source data 4.** Source data for *Figure 2E*.
**Figure supplement 1.** Range of speeds by sample.
**Figure 2—video 1.** Heterogeneity of T cell migration.
https://elifesciences.org/articles/53933#fig2video1

To evaluate the rate of speed switching in our system, we measured the average speeds of individual trajectories on non-overlapping ~20 min intervals, and evaluated how the speed ranks change as a function of the time between intervals (*Figure 2B*). We found a high correlation between speeds on adjacent non-overlapping intervals, which decays slowly on the timescale of the measurement. Thus each cell samples a characteristic distribution of speeds that is stable over one to two hours. For the remainder of the analysis, we will consider the average speed to be a property of the trajectory; we return to consider the implications of speed switching in the discussion.

We note that we observed variation in the distributions of cell speeds between samples: overall, 48% of the variance in cell speeds can be explained by the sample identity. Nonetheless, the distributions of cell speeds within each sample are broad and overlapping (*Figure 2—figure supplement 1*), accounting for the majority of the variance (52%). Amongst other effects, sample to sample variation could be the result of differences in antigen environment or global cytokine levels between fish.

## Heterogeneous cell migration statistics fall on a behavioral manifold

Previous work (*Maiuri et al., 2015*) has suggested that actin flows may generate a coupling between speed and directional persistence in migrating cells. This study generates the hypothesis that cells, in general, are not free to pick any turn and speed statistics, but rather that there may be underlying biophysical constraints. To investigate this hypothesis in our system, we divided the cells into quintiles based on speed, which we refer to as speed classes. We observed strong variation in the distribution of turn angles amongst speed classes (*Figure 3A*): fast cells are most likely to turn shallowly, slow cells are most likely to turn around, and the distribution varies smoothly across the speed classes. This dependence could be driven by a local coupling between speed and turn angle: cells tend to go straighter whenever they go fast, which the faster cells do more often. Alternatively, it could be driven by an overall behavioral difference between fast and slow cells. To distinguish these possibilities, we measured the average turn angle as a function of the size of the steps surrounding it (*Figure 3B*). We found that both of these effects contribute: all cells go straighter during faster periods, but for a given step size, slow cells are more likely to turn sharply.

The relationship between speed and turning suggests that there may also be systematic differences in the scaling of the MSD at short times between cells. In particular, variation in speed alone amongst individuals would not change the shape of the MSD, which would collapse when appropriately scaled (Appendix 1). On the other hand, the systematically shallower turns of faster cells would be expected to boost the slope of their MSD at short times, an effect we observe in the data (*Figure 3C*).

The analysis at the level of speed classes suggested that there might be a single scalar variable, for which the cell's average speed is a good proxy, that determines a number of higher-order statistics characterizing the cell's migration behavior. To test this at the level of individual trajectories, we chose two summary statistics that capture the cell's turning behavior: the average of the cosine of the turn angles along the trajectory, and the correlation between speeds and turn angles along the trajectory. The former is a summary of the overall distribution of turn angles for that cell, while the latter captures the degree of additional local coupling between speed and turn angle. Together with the cell's speed, these two summary statistics form a three-dimensional behavioral space. We observed that the cell trajectories fall close to a curve in this space (*Figure 3D*). In particular, 73% of the variance in the average cosine can be explained by cell speed, with some residual variance due to the stochasticity of the process (7%) and other unknown effects (20%) (*Figure 3—figure supplement 1*). Thus T cell migration statistics can be organized into a one-dimensional behavioral manifold, characterized by a strong dependence between speed and turning behavior.

## Model predicts wide variation in length scales of exploration across the population

Our observation of a behavioral manifold suggests that, despite the apparent heterogeneity in migration strategies, there may be a common program with a single underlying variable, consistent with the work of Mauri et al. In this view, a cell's location on the manifold reflects its internal value of this control variable, which in turn dictates its random walk behavior. Given the results of our MSD analysis, to determine candidates for a single-parameter migration model, we started with the canonical persistent random walk (Ornstein-Uhlenbeck) process (*Uhlenbeck and Ornstein, 1930*):

$$\frac{dv_i}{dt} = -\frac{1}{P}v_i + \frac{S}{\sqrt{P}}\eta, \tag{1}$$

where v is the velocity, $\eta$ is a white noise term, and $i$ labels the velocity component. This model has two free parameters: the speed, S, and the persistence time, P, which is the average time before a cell turns. (Note that speeds inherently vary along trajectories in this model; S controls the average speed.) Our observations suggest that there may in fact only be one control parameter; in particular, because faster cells tend to make shallower turns, we expect P to increase with S. To determine the relationship between these two variables, we measured the persistence time, averaged along each trajectory, as a function of cell speed, and found a linear dependence (*Figure 4A*). This suggests the following simple model of cell motility:

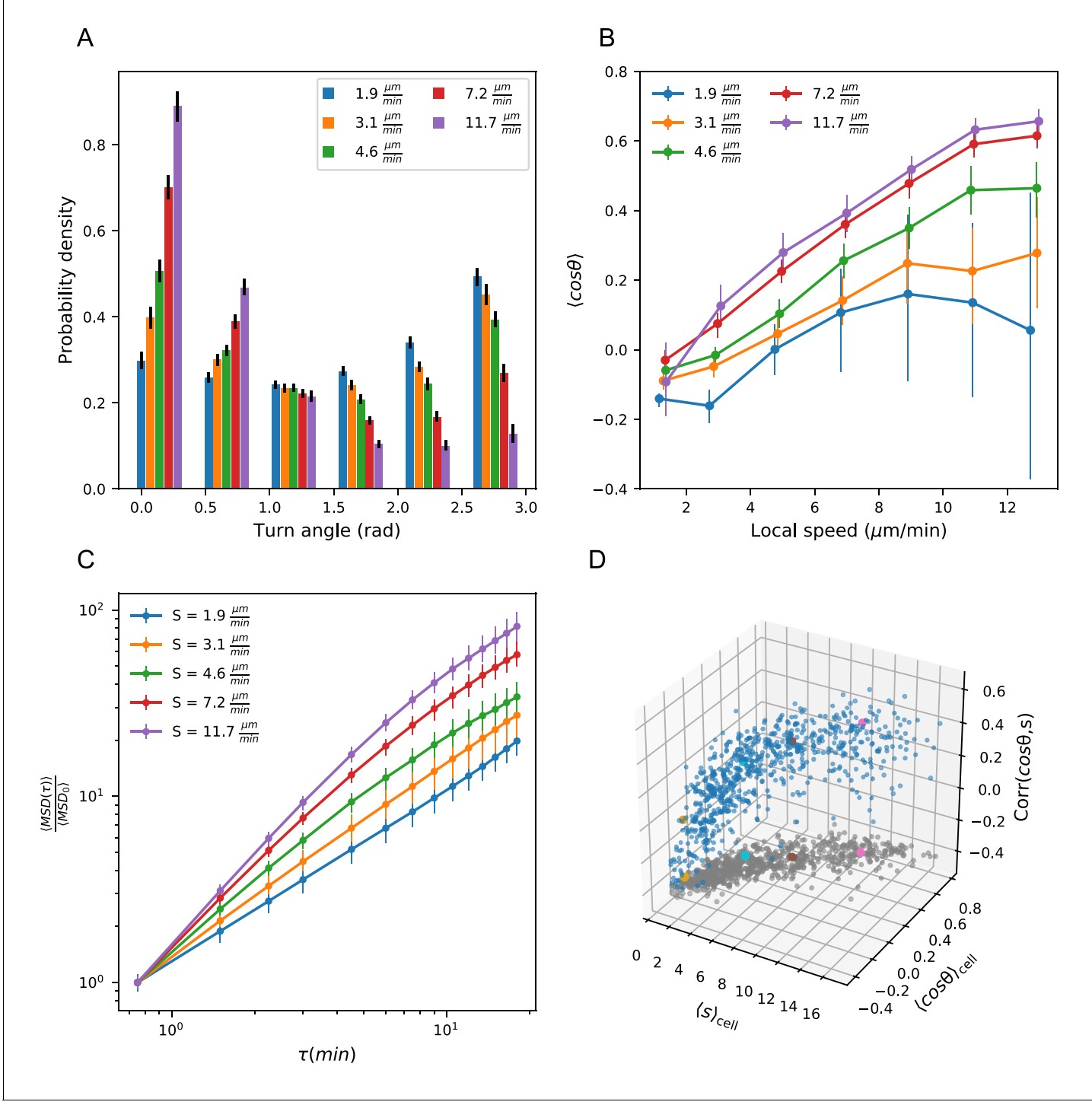

**Figure 3.** Heterogeneous cell migration statistics fall on a behavioral manifold. (**A**) Distribution of turn angles amongst cells grouped by speed class. The distribution varies smoothly from faster cells, which tend to go straighter, to slower cells, which tend to turn around more often. Error bars represent 95% confidence intervals from a bootstrap over trajectories in each speed class. The legend reports the mean speed for trajectories in each class. (**B**) Turning behavior conditioned on current cell speed. The average of the cosine of the turn angle as a function of the average length of the steps on either side. Cells are grouped into speed classes as in A. Error bars represent 95% confidence intervals from a bootstrap over trajectories in each speed class. (**C**) Mean squared displacement by speed class. Due to the variation in turning behavior, the faster cells appear initially more superdiffusive. Error bars represent 95% confidence intervals from a bootstrap over trajectories in each speed class. All speed class calculations were performed on the n = 569 trajectories that included all time intervals in the MSD analysis. (**D**) Organization of cell behavior into a curve in a three

*Figure 3 continued on next page*

*Figure 3 continued*

dimensional behavioral space. Each point represents a trajectory, and we show the average speed, turn angle, and local speed-turn correlation. Grey: projection into the x-y plane. The trajectories shown in *Figure 2* are colored. Trajectories pooled over n = 16 fish.

The online version of this article includes the following source data and figure supplement(s) for figure 3:

**Source data 1.** Source data for *Figure 3A*.
**Source data 2.** Source data for *Figure 3B*.
**Source data 3.** Source data for *Figure 3C*.
**Source data 4.** Source data for *Figure 3D*.
**Figure supplement 1.** Variance explained by speed-turn relationship.

$$\frac{dv_i}{dt} = -\frac{1}{\frac{S}{\alpha}+\beta}v_i + \frac{S}{\sqrt{\frac{S}{\alpha}+\beta}}\eta \tag{2}$$

where $\alpha$ is a constant with units of acceleration, $\beta$ is a constant with units of time, and both are constrained by the empirical relationship in *Figure 4A*. We call this the speed-persistence coupling model (SPC).

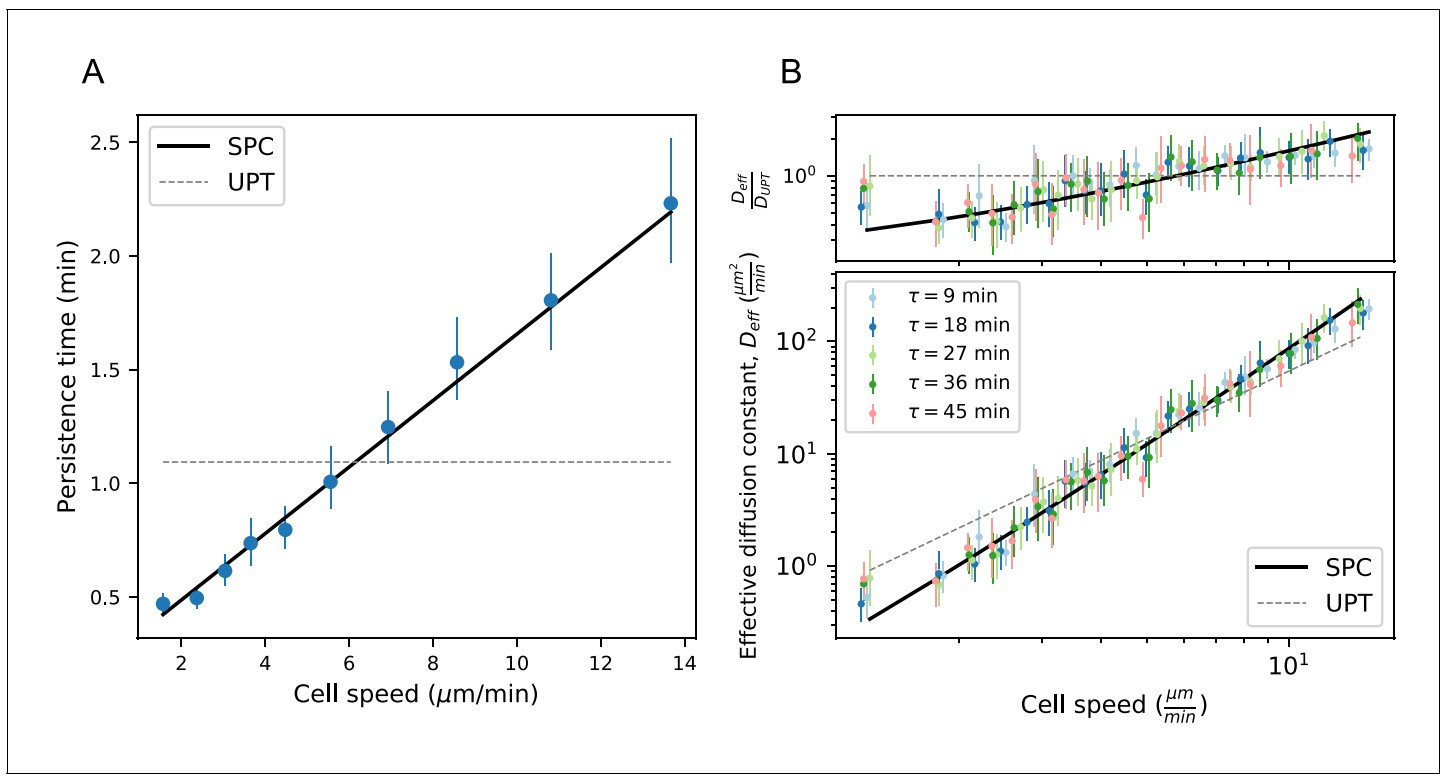

**Figure 4.** Model predicts wide variation in length scales of exploration across the population. (A) Mean persistence time as a function of cell speed, measured along trajectories (n = 710). Error bars represent 95% confidence intervals from a bootstrap over trajectories. UPT: Uniform persistence time; SPC: Speed-persistence coupling. (B) Scaling of the effective diffusion constant with cell speed. Except for a constant offset, parameters are fixed based on the speed-persistence relationship in A. Error bars represent 95% confidence intervals on a bootstrap over trajectories. Numbers of trajectories in each time interval: n = 704; n = 654; n = 607; n = 558; n = 523. Trajectories were pooled over n = 16 fish.

The online version of this article includes the following source data and figure supplement(s) for figure 4:

**Source data 1.** Source data for *Figure 4A*.
**Source data 2.** Source data for *Figure 4B*.
**Figure supplement 1.** Statistics from *Figures 3* and *4*, with timepoints subsampled.
**Figure supplement 2.** Speed-persistence relationship with shallower cut-off angle.

As in other persistent random walk models, SPC walkers are diffusive at long times; the MSD scales linearly with time, and the ratio between these quantities defines an effective diffusion constant (Appendix 1):

$$D_{eff} \equiv \frac{MSD(\tau)}{4\tau} = \frac{1}{2}S^2 P. \qquad (3)$$

Due to the dependence of P on S, the SPC model predicts a strong scaling of the effective diffusion constant with cell speed. We tested this prediction at several time intervals $\tau$ and found good quantitative agreement between the model and the data (*Figure 4B*). In particular, heterogeneity in speeds creates a broad range of effective diffusion constants, spanning 2 orders of magnitude. The coupling between speed and persistence amplifies this effect, generating five-fold more variation in the effective diffusion constants across the cells than would be expected for a uniform persistence time model (UPT). We also note that the collapse of $D_{eff}$ measured at different time lags $\tau$ provides additional corroboration of diffusive (as opposed to Levy/super-diffusive) scaling.

The analyses in this and the previous section depend on measured cell speeds and turn angles, which are an imperfect proxy for the true instantaneous process (*Beltman et al., 2009b*). In particular, both noise in the cell locations and finite sampling intervals can introduce bias in the measured speeds, which could in principle generate spurious relationships between measured speed and turning behavior. We took two approaches to addressing the sensitivity of our conclusions to these issues. First, we addressed sensitivity to sampling rate by repeating the analyses above, subsampling timepoints by a factor of 2. This makes the turning behavior of the slowest two speed classes harder to distinguish, because they are rarely persistent over more than one timestep (*Figure 4—figure supplement 1A,D*), and introduces more noise in the local coupling and persistence relationships (*Figure 4—figure supplement 1B–C*), but otherwise does not alter the structure of the correlations (*Figure 4—figure supplement 1A–F*). Second, we assessed the potential biases introduced by mislocation noise and finite sampling to the speed-persistence relationship in simulations (Appendix 1, *Appendix 1—figure 1*). We found that mislocation noise can lead to spurious correlations between speed and persistence at the slow end of the speed spectrum, but cannot account for the consistent correlation we observe across speeds.

Additionally, we note that our measured trajectories stay predominantly within the tail fin and larval fin fold (*Figure 1—figure supplement 1*), suggesting a boundary between the fin fold and muscle region of the tail. Such a boundary could influence the MSD. However, we compared to the MSD calculated on one held-out sample not subject to these boundary effects and observed no difference (see *Appendix 1—figure 2*).

Finally, we note that the SPC Langevin model describes the effective diffusive behavior of the trajectories and their scaling at longer times, but may not capture all the details of the microscopic dynamics. In particular, the propensity of trajectories to turn backwards (peak at $\theta = \pi$ radians, *Figure 3A*) is not captured by this model.

## Manifold is preserved under a drug perturbation to cell speeds, and in mouse T cells and *Dictyostelium*

We next asked about the robustness of the observed behavioral manifold under a perturbation to cell speeds. Given the role of actin nucleation and remodeling in leukocyte motility (*Vicente-Manzanares et al., 2002*), we chose the drug Rockout, a known Rho kinase inhibitor affecting this pathway (*Barros-Becker et al., 2017*), as a candidate for perturbing cell speed, and repeated the measurements and analysis of cell migration behavior in the presence of the drug (Materials and methods). We found that the distribution of cell speeds shifted downwards, but we still observed a quantitatively similar positive relationship between speed and turning behavior (*Figure 5A,B*). This is consistent with a model where the perturbation primarily shifted an internal cell state variable that determines location along the behavioral manifold, which in turn dictates both speed and turning behavior, although we note that there may be an additional small shift towards shallower turns in the drug condition.

Finally, we analyzed published data from two other species, mouse T cells in situ (*Gérard et al., 2014*) and *Dictyostelium* (*Dang et al., 2013*), in this framework. While some of the analyses that depend on longer time traces and larger cell numbers are not possible with these datasets, we

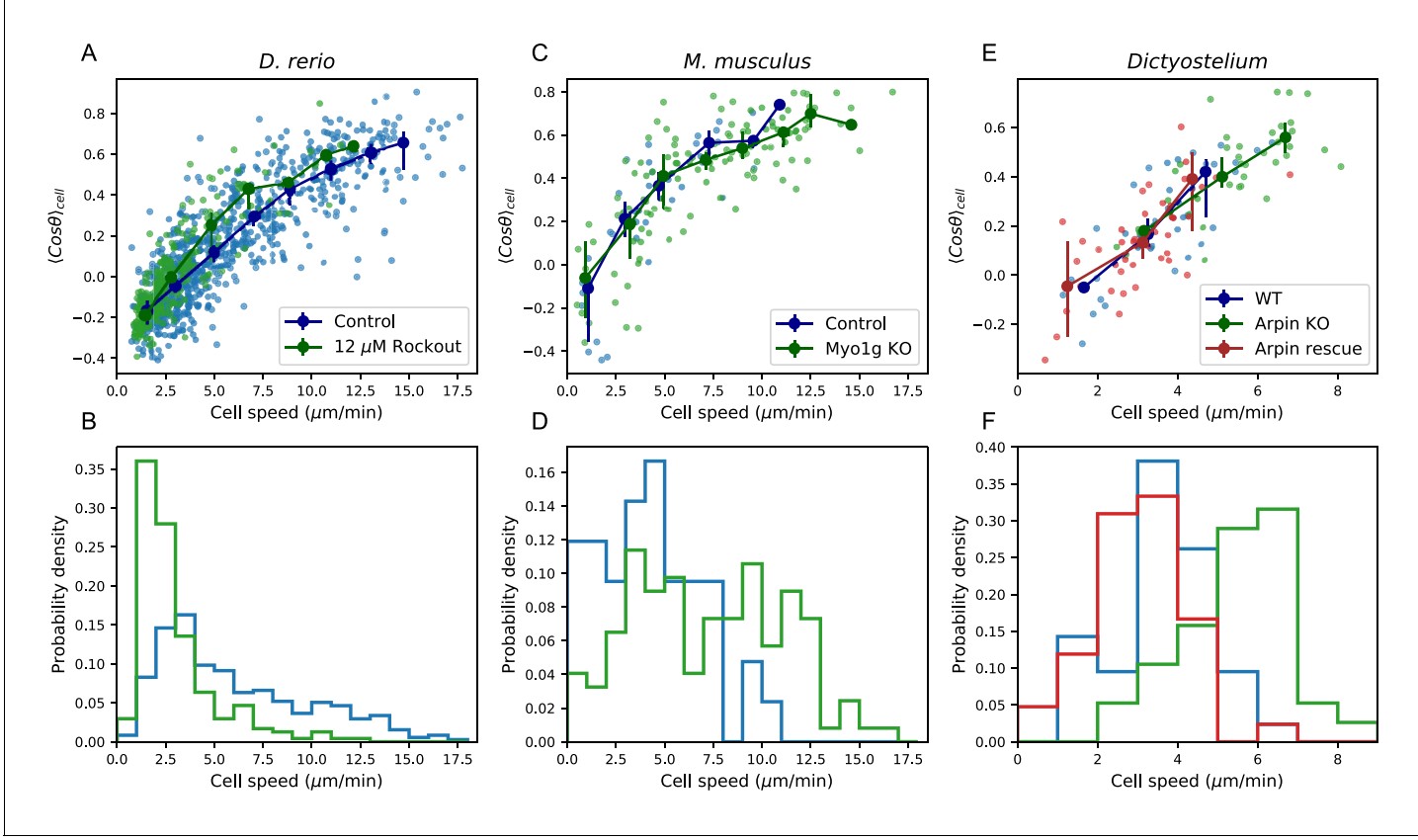

**Figure 5.** Manifold is preserved under a drug perturbation to cell speeds, and in other species. (**A**) Correlation between the average cosine of the turn angles along the trajectory and cell speed, for cells in control and Rockout-treatment conditions. Data for all cells is shown as well as a binned average. Error bars represent 95% confidence intervals on the binned average on a bootstrap over cells. (**B**) The distribution of speeds amongst control and Rockout-treated trajectories. The treatment lowers cell speeds but maintains the relationship between speed and persistence. Statistics based on trajectories pooled over n = 16 control fish (n = 712 trajectories) and n = 6 Rockout treatment fish (n = 236 trajectories). (See also *Figure 5—figure supplement 1*. (**C**) As in A, for mouse T cells (data from *Gérard et al., 2014*). The perturbation is a genetic knockout of a non-canonical myosin motor, *Myo1g*. E. As in A, for *Dictyostelium* (data from *Dang et al., 2013*). The perturbations are a knockout and rescue of the *Arp2/3* inhibitor Arpin. (control: n = 42; *Myo1g* KO: n = 123) (**D,F**) Distributions of cell speeds for the control and treatment conditions shown in C,E. (WT: n = 42; Arpin KO: n = 38; Arpin rescue: n = 42) In each case, the distribution of speeds shifts, but the cells tend to move along the speed-turn curve. ).

The online version of this article includes the following source data and figure supplement(s) for figure 5:

**Source data 1.** Source data for *Figure 5A*.
**Source data 2.** Source data for *Figure 5B*.
**Source data 3.** Source data for *Figure 5C*.
**Source data 4.** Source data for *Figure 5D*.
**Source data 5.** Source data for *Figure 5E*.
**Source data 6.** Source data for *Figure 5F*.
**Figure supplement 1.** *Figure 5A,B*, including only paired control-Rockout treatment samples.

tested the relationship between average turn angle and cell speed. We found that this correlation held amongst the control cells in both studies (*Figure 5C–F*). This suggests that, as for zebrafish T cells, there is heterogeneity in speed and turning behavior amongst the cells, and is consistent with a similar behavioral manifold. In the two published studies, genetic perturbations that knocked out or down one member of the actin remodeling machinery were used: a knockout of the non-canonical myosin *Myo1g* in one case, and a knockout of the *Arp2/3* inhibitor Arpin in the other. In each case, the perturbation had a substantial effect on the distribution of cell speeds (*Figure 5*, E-F). However, in both cases, a quantitatively similar positive relationship between the speed and turning behavior amongst the perturbed cells was preserved.

## Single cell RNA sequencing suggests transcriptional heterogeneity in actin nucleation activity

Our observation that cells maintain characteristic distributions of speeds over periods of a few hours suggests that there may be variation in transcriptional state that underlies some of the heterogeneity in cell migration behavior. To investigate this possibility, we performed single-cell RNA sequencing on cells isolated from the tail of 15 dpf Tg(*lck*:GFP) zebrafish. To assess the fidelity of the marker, we sorted GFP+ cells from an unbiased FSC/BSC gate (Materials and methods). We used standard dimensional reduction and clustering methods (Materials and methods) to identify 330 putative T cells (*Figure 6—figure supplement 1*). Unexpectedly, we also identified a population of epithelial cells that may mis-express *lck* at low levels (Materials and methods, *Figure 6—figure supplement 1*, *Figure 6—figure supplement 2*).

We next used a self-assembling manifold algorithm designed to detect subtle variation (*Tarashansky et al., 2019*) to examine finer-scale structure within the putative T cell cluster. This algorithm revealed a plate effect related to one of our sort plates (*Figure 6—figure supplement 3*), which we therefore excluded from the remainder of the analysis. We repeated the SAM analysis on the remaining (n = 237) cells, and identified two main subtypes (*Figure 6A–B*), consistent with a previous report (*Tang et al., 2017*). We note that the prior study identified the smaller subpopulation as NK cells; however, in addition to the previously-reported marker genes, we find that these cells have moderate expression of the T cell receptor *trac*. We have therefore chosen not to annotate these as NK cells.

Both subpopulations of cells express high levels of genes canonically involved in actin nucleation and remodeling in the leukocyte cytoskeleton (*Vicente-Manzanares et al., 2002*; *Takenawa and Suetsugu, 2007*) (WASP/ARP2/3 pathway; *Figure 6B*); indeed, these genes are amongst the top differentially expressed between the T cells and putative epithelial cells in our sample (*Figure 6—figure supplement 1*, *Figure 6—figure supplement 2*). To analyze variation in this pathway amongst the T cells, we chose *arpc1b*, a subunit of the ARP2/3 complex, as a reference gene because ARP2/3 directly nucleates actin during ameboid cell migration, and its inhibition is known to modulate cell speed (*Dang et al., 2013*). We tested the rank correlation of expression of all other moderate to high expressed genes with *arpc1b* (*Figure 6C*). We found that 3 of the four top correlated genes (those that are statistically significant after Bonferonni correction) are also canonically involved in this pathway: the actin monomer genes *act2b* and *act1b*, as well as the upstream activator *cdc42l*. Unexpectedly, the fourth gene encodes a lincRNA whose biological function is unknown. Thus we detect real, although not strong, co-variation in genes involved with actin nucleation, which may create long-lived cell-intrinsic variation in motility.

While the two T cell subpopulations separate in some dimensions of gene expression space, including, for example, the marker genes shown in *Figure 6B*, both groups express *arpc1b* and its correlates. Indeed, the distributions of expression for the two subtypes overlap for these genes (*Figure 6D*). This is consistent with a single continuous motility axis, without two distinct subgroups, as in our microscopy data.

We have observed transcriptional heterogeneity in actin nucleation genes, which may underly variability in motility states. This observation suggests that cells may vary along a 'nucleation high' to 'nucleation low' axis, generating a range of long-lived speed states. However, it remains technologically infeasible to directly associate a cell trajectory with a gene expression profile, so a direct test awaits future work. Additionally, we note that regulation that occurs at the protein level, for example through phosphorylation states, is likely very important to shorter timescale variation in cell motility behavior, and is not reflected in gene expression.

## Discussion

We have measured and analyzed the variability in cell motility amongst the T cells of the zebrafish tail. We found that cell motility statistics are inconsistent with Levy flight; rather, speeds are heterogeneous from cell to cell. We note that a previous study that reported modified Levy flight in T cells (*Harris et al., 2012*) was carried out in a very different biological context (adoptively transferred CD8+ T cells in mouse CNS in the presence of an infection), which could drive differences in motility behavior. However, we note that the statistics of our trajectories closely resemble those measured by Harris et al. up through the 10 min time lag analyzed in that study.

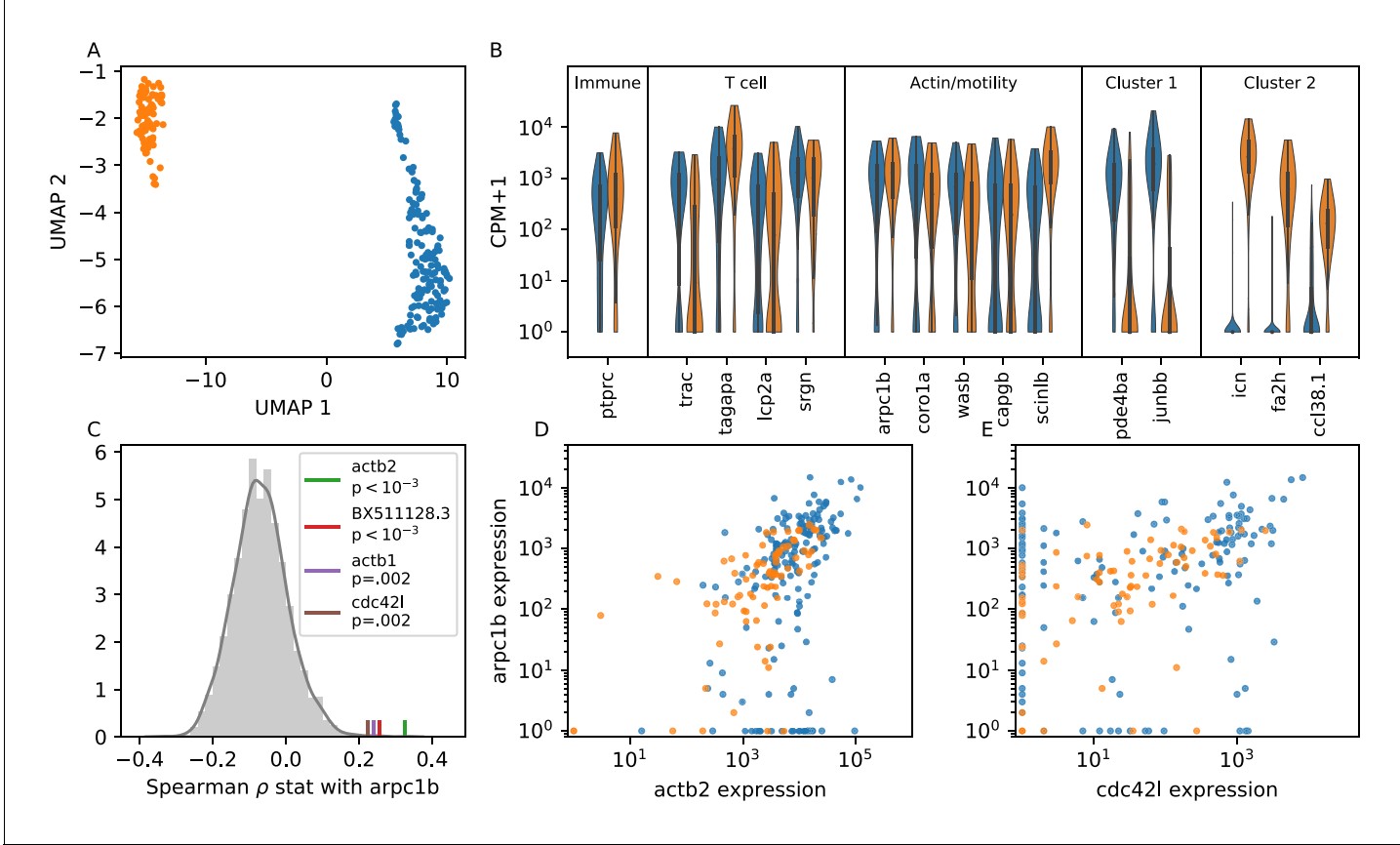

**Figure 6.** Single cell RNA sequencing shows moderate covariation in actin nucleation across T cells. (**A**) UMAP dimensional reduction of single-cell RNA sequencing profiles of zebrafish T cells (Materials and methods) shows two main subtypes. Cluster colors are shared across panels **A, B, D, and E**. For a list of differentially-expressed genes between clusters, see *Supplementary file 1*. (**B**) Violin plots of marker genes in common to both subtypes (including immune, T cell, and motility markers), as well as selected marker genes for each subtype. (For the list of differentially-expressed genes, see *Supplementary file 1*). (**C**) Distribution of a rank correlation coefficient-related statistic between *arpc1b* and other moderate and high expressed genes, amongst all T cells (Materials and methods). Statistically significant outliers (after Bonferonni correction) are colored and labeled. The top correlated genes include three also involved in actin nucleation activity. (**D**) Correlation between expression (counts + 1) of *arpc1b* and *actb2* across the T cells, with colors as in A. (**E**) Correlation between expression (counts + 1) of *arpc1b* and *cdc24l*, with colors as in A. Analysis performed on n = 237 cells (see Main Text, Materials and methods).

The online version of this article includes the following source data and figure supplement(s) for figure 6:

**Source data 1.** Source data for *Figure 6A*.
**Source data 2.** Source data for *Figure 6B*.
**Source data 3.** Source data for *Figure 6C*.
**Source data 4.** Source data for *Figure 6D*.
**Source data 5.** Source data for *Figure 6E*.
**Figure supplement 1.** Comparison between UMAP and index sort, over all cells.
**Figure supplement 2.** Differential expression between the two main clusters.
**Figure supplement 3.** Plate effect in variation within T cell cluster.

We also found that migration statistics from zebrafish T cells as well as mouse T cells and *Dictyostelium* fell on a behavioral manifold, characterized by a coupling between speed and directional persistence. We note that, in general, heterogeneity of motility behavior across a population could be caused by the tissue context rather than by cell-intrinsic factors. However, the effects of the drug perturbation, as well as the effects of the genetic perturbations from *Gérard et al., 2014* and *Dang et al., 2013*, support a cell-intrinsic basis for the behavioral manifold we observe here. In particular, we performed trials in which the same regions of tissue were imaged and cells tracked before and after addition of the drug (*Figure 5—figure supplement 1*); the observed changes in migration

statistics must then be caused by the drug's effect on the cell's internal state, not by the tissue context.

The drug perturbation experiment and analysis of data from the other species suggests that there is one underlying cell-intrinsic variable that jointly controls speed and directional persistence. Maiuri et al. also observed a coupling between speed and directional persistence across multiple cell types, and performed elegant in vitro work to demonstrate that actin retrograde flow speed is correlated both with cell speed and persistence time (*Maiuri et al., 2015*). Using single cell RNA sequencing, we observed covariation amongst T cells in a group of genes involved in actin nucleation. This suggests that cells vary transcriptionally in their actin nucleation activity. Levels of these genes may represent an underlying cell control variable that determines actin retrograde flow speeds and hence both cell speed and persistence. A direct test of this hypothesis awaits development of techniques to link a single cell's trajectory and its gene expression profile.

In two previous studies, genetic perturbations were performed that made cells faster and more persistent on average (*Gérard et al., 2014*; *Dang et al., 2013*). Our results suggest that this connection may be general to the cells rather than specific to the perturbation. In particular, shifts in the average turning behavior have been used to argue that Arpin and Myo1g control cell steering. Our analysis suggests that increasing cell speed may in many cases increase straightness, and vice versa, so that the effect on cell steering may be indirect.

We have analyzed cell migration in the context of a speed-persistence coupling model, which is a modified Orenstein-Uhlenbeck model where the speed and persistence time parameters are explicitly co-dependent. We chose this as the minimal model that captures both the diffusive behavior at long times and the coupling between speed and persistence, allowing for a prediction of the range of effective diffusion constants, and hence length scales of exploration, amongst the cells (*Figure 4*). Another feature of measured trajectories, in previous studies and in our work, is variation in speed along the trajectory, sometimes referred to as intermittency. We note that the Langevin dynamics of the SPC model inherently generate variation in speed, including pauses, along trajectories: the speed parameter sets the distribution of instantaneous speeds sampled by the cell. Other approaches have included adding pauses to a model with a fixed distribution of step sizes (*Harris et al., 2012*). Maiuri et al. considered an explicitly active model where actin dynamics within the cell generated motility. While slightly more complex, this model has several interesting features, including that the cell motility emerges from internal biophysical mechanisms, and also that it produces multiple phases of migratory behavior, including one with additional intermittency (*Maiuri et al., 2015*). Future work could use transgenic methods to label actin in the zebrafish to directly test this active matter model in vivo. Another mechanistic basis for intermittency was demonstrated by *Dong et al., 2017*, who showed that spontaneous cytosolic calcium signaling generates pauses during basal T cell motility. Investigation of this mechanism and its potential role in producing the speed fluctuations observed in our system would also be an interesting subject for further work.

Our results show that across the population, cells explore over a very broad range of length scales, covering orders of magnitude of variation in effective diffusion constants. This variability is driven primarily by differences in average speed between cells, which is amplified by the observed coupling between speed and persistence. However, analysis of the effectiveness of this variation as a search strategy, as compared with modified Levy flight (*Harris et al., 2012*) or models with additional intermittency (*Maiuri et al., 2015*), would require a detailed knowledge of the distribution of targets in the tissue as well as an additional order of magnitude in time of observation of the cell trajectories in our system, to fully characterize the slow timescale speed switching. We additionally note that several studies (*Dustin et al., 1997*; *Mempel et al., 2004*; *Kawakami et al., 2005*; *Castellino et al., 2006*; *Moreau et al., 2015*; *Dong et al., 2017*; *Negulescu et al., 1996*) have indicated that T cells react to local chemical signals by changing speed–in particular, that calcium signaling can cause cell arrest, and this is important to contact with antigen presenting cells and antigen recognition. This suggests that the local signaling environment may also be important to shifting the cell behavior along the manifold. Detecting these signaling and recognition events in vivo in real time would be an exciting avenue for future work.

# Materials and methods

**Key resources table**

| Reagent type (species) or resource | Designation | Source or reference | Identifiers | Additional information |
|---|---|---|---|---|
| Strain, strain background (*Danio rerio*) | Tg(*lck:GFP*) | Gift from Aya Ludin-Tal and Leonard Zon *Langenau et al., 2004* | | |
| Chemical compound, drug | Tricaine-S, MS-222 | Pentair | | |
| Chemical compound, drug | Rockout | Sigma Aldrich | #555553 | |
| Software, algorithm | Ilastik | *Sommer et al., 2011* | | |
| Software, algorithm | Custom analysis software | This paper | | Available at https:// github.com/ erjerison/ TCellMigration |
| Software, algorithm | STAR | *Dobin et al., 2013* | | |
| Software, algorithm | htseq | *Anders et al., 2015* | | |
| Software, algorithm | SAMalg | *Tarashansky et al., 2019* | | |

## Zebrafish lines and procedures

Tg(*lck*:GFP, *roy*^-/-, *nacre*^-/-) zebrafish (*Danio rerio*) (*Langenau et al., 2004*) were obtained as a generous gift from Dr. Leonard Zon and Dr. Aya Ludin-Tal. Imaging was performed on Tg(*lck*:GFP) zebrafish crossed into a *nacre*^-/- background, at between 9 and 13 dpf. All adult and larval zebrafish were maintained according to protocols approved by the Stanford Administrative Panel on Laboratory Animal Care.

## Microscopy

Imaging was performed on a single-plane illumination microscope constructed as specified in *Pitrone et al., 2013*, with the exception that a Prior ProScan XY stage (Prior Scientific) coupled to a Zaber T-LLS 105 stage (Zaber Technologies) was used for sample movement. The light sheet was generated using an Olympus UMPLFLN10XW objective (NA = 0.3) and detection was performed with an Olympus UMPLFLN20XW objective (NA = 0.5) and an achromatic doublet tube lens (AC508-180-A-ML, Thorlabs). Images were recorded either on a Retiga 2000R camera (Qimaging) or an Ace acA2040 (Basler). For the Ace ac2040 camera, a meniscus lens (LE1418-A - O2' N-BK7, Thorlabs) was added as a zoom lens, to match the image pixel width between the two cameras at $.37\,\mu m$. The fluorescence source was an Obis LS 488 nm laser (Coherent), and the microscope was controlled by Micro-Manager.

Zebrafish between 9 and 13 dpf were anesthetized with Tricaine-S (MS-222, Pentair; .008% w/v, buffered to pH 7) and embedded in 2% low melting point agarose (Lonza SeaPlaque, #50100) with .004% w/v Tricaine. For imaging, the agarose was submerged in E3 with .008% w/v Tricaine and 50 mM Hepes. With the exception of *Figure 2* and *Figure 2—video 1*, tiled z-stacks were obtained every 45 s for at least 180 timepoints, with a field of view of at least 592 $\mu m$ (dorsal-ventral axis) by 1200 $\mu m$ (anterior-posterior axis), in the tail region posterior to the anus. For *Figure 2A* and *Figure 2—video 1*, a z-stack was obtained every 12 s for 1100 timepoints, with a field of view of 757 × 568 µm (the first 900 timepoints are shown). For statistical comparison with the remainder of the data, trajectories from this final dataset were subsampled in time to give 48 s timesteps. Data was acquired with 2 × 2 binning, for an image pixel width of .74 µm.

For imaging in the presence of Rockout, embedded fish were submerged in E3 with .008% w/v Tricaine and 50 mM Hepes plus 12 µM Rockout (Sigma Aldrich #555553). For paired control/Rockout trials, fish were imaged for 2.5 hr in control conditions, followed by 2.5 hr in Rockout conditions over the same field of view.

## Single-cell RNA sequencing

Thirty 15 dpf Tg(*lck*:GFP) zebrafish were euthanized using .04% w/v Tricaine and transected posterior to the anus. Tail portions were pooled into HBSS (ThermoFisher #14025092) on ice. Tails were dissociated by incubating with 100 $\mu g/mL$ Liberase-TL (Sigma Aldrich #5401020001) at room temperature for 20 min, followed by trituration with a 23 gauge needle. The cell suspension was filtered through a 40 $\mu m$ filter and washed once in HBSS. GFP+ cells were sorted from an unbiased FSC-SSC gate on a Sony SH800 cell sorter into 384-well hard-shell PCR plates (Bio-Rad HSP3901) containing .4 µl of lysis buffer, prepared as described previously (*Tabula Muris Consortium et al., 2018*). Reverse transcription following a Smart-Seq2 protocol, and Illumina library preparation, were carried out as described previously (*Tabula Muris Consortium et al., 2018*), except that following cDNA amplification, cDNA was diluted uniformly to a mean target concentration of .4 $ng/\mu l$ for library preparation. Libraries were sequenced on the NovaSeq 6000 Sequencing System (Illumina) using 2 × 100 bp paired-end reads.

## Image processing and cell tracking

Tiles were assembled based on recorded stage coordinates and a Maximum Z projection was applied to Z stacks. Sample drift in x and y was subtracted by identifying and tracking autofluorescent pigment spots. In particular, the coordinates of 1–3 isolated pigment spots were identified manually at the first timestep; at each timestep, the brightness centroid was computed for a circle with a 25 pixel radius around the previous centroid, and the average trajectory of the pixel spots was rounded to the nearest pixel and subtracted from the timeseries. Prior to cell segmentation, the average image across the whole timecourse was subtracted from each timestep. For data recorded on the Retiga 2000R camera, prior to segmentation the image was thresholded at the 30th pixel percentile and the maximum pixel value was fixed so that .4% of pixels were saturated. For data recorded on the Basler Ace acA2040 camera, no lower threshold was used and the maximum pixel value at each timepoint was fixed so that .2% of pixels were saturated. Ilastik software (*Sommer et al., 2011*) was used for cell segmentation and tracking: the Ilastik pixel classification module was used to classify foreground and background, and the manual tracking module was used to identify and track cells. Tracks were terminated if two segmented cells collided and it was not possible to disambiguate, or if a segmented cell was lost due to passing into an autofluorescent region or due to imperfections in segmentation and/or illumination. To define trajectories, the brightness centroid of each cell in x and y at each timestep was computed from Ilastik tracking masks and the Maximum Z projection. Processing steps not using Ilastik were performed using Python 3.6 (code available at: https://github.com/erjerison/TCellMigration; *Jerison, 2020*; copy archived at https://github.com/elifesciences-publications/TCellMigration).

## Trajectory analysis

Trajectories with at least 30 consecutive steps were included in the analysis; for MSD calculations, trajectories that included all time intervals were included. For calculations of power spectra, single missing timesteps were linearly interpolated based on the two adjacent positions, and computations were performed on the longest consecutive segment for each trajectory. For the *M. musculum* data, the time interval was 30 s. For the *Dictyostelium* data, timesteps were subsampled from the original to give an interval of 20 s. Unless otherwise noted, statistics are reported as the median of the statistic on a bootstrap over trajectories.

Mean-squared displacements were computed along each trajectory as:

$$MSD(\tau = mt_{int}) = \frac{1}{N-m}\sum_{s=1}^{N-m} ||\vec{x}(m+s) - \vec{x}(s)||^2, \tag{4}$$

where $N$ is the total number of timesteps and $t_{int}$ is the time interval. The overall MSD was computed by averaging the MSDs for each trajectory, and 95% confidence intervals were calculated via a bootstrap over trajectories.

The overall MSD was fit to:

$$MSD(\tau) = 4S^2 P\tau(1 + \frac{P}{\tau}(e^{-\frac{\tau}{P}} - 1)) + \sigma^2, \tag{5}$$

which we note is the common formula for mean squared displacement in both the Ornstein-Uhelnbeck model (see Appendix 1) and in the Kratky-Porod wormlike chain model. Unless otherwise noted, fitting was performed using the scipy.optimize.curvefit function in scipy 1.3.0; fitting was performed in log space and weighted by computed confidence intervals.

The velocity power spectrum was computed based on the vector of secant-approximated velocities for each trajectory. Velocity vectors were zero-padded to 400 timesteps, and the fourier transforms of the velocity components were computed using the the fft function in numpy (1.16.4). Letting the fourier-transformed velocity components for trajectory $m$ be $v_x(k,m)$, $v_y(k,m)$, the power spectrum for each trajectory was computed as:

$$PSD(k,m) = \frac{1}{N^2}\frac{N}{N_m}\sum_{i=x,y}(||v_i(k,m)||^2 + ||v_i(N-k,m)||^2), \ 1<k<\frac{N}{2} \tag{6}$$

$$PSD(0,m) = \frac{1}{N^2}\frac{N}{N_m}\sum_{i=x,y}||v_i(0,m)||^2 \tag{7}$$

$$PSD(\frac{N}{2},m) = \frac{1}{N^2}\frac{N}{N_m}\sum_{i=x,y}||v_i(\frac{N}{2},m)||^2, \tag{8}$$

where $N=400$ and $N_m$ is the length of trajectory $m$. The overall PSD was computed as the average over the PSDs for each trajectory:

$$PSD(k) = \frac{1}{n}\sum_{m=1}^{n}PSD(k,m), \tag{9}$$

where $n$ is the number of trajectories. For *Figure 1D*, a piecewise linear function was fit to the PSD in log space; we plot the high-frequency fitted line and a line with slope 0.

Following (*Petrovskii et al., 2011*), we calculated the distribution of bout lengths within a trajectory as the distribution of x displacements between reversals in direction in x, divided by the average of these displacements within each trajectory. The distribution was calculated using the numpy.histogram function on percentile bins with the option density = True; the x locations of points were determined based on the average value of points in each bin. We fit the distribution to a stretched exponential function $f(x) = Ae^{-\gamma x^\beta}$; the fitted value of the stretch parameter $\beta$ was .9.

The overall speed distribution was computed by collecting secant-approximated speeds across all trajectories and timepoints; similarly, the overall turn angle distribution was computed by collecting all relative angles between consecutive segments. For the Kolmogorov-Smirnov (KS) test, the overall CDFs of speeds and turn angles were estimated by measuring the cumulative frequency over 25 percentile bins and performing linear interpolation to yield a continuous function. A two-sided KS test (scipy.stats.kstest) was performed for the sets of speeds and turn angles of each trajectory.

We evaluated the fraction of the variance in cell speeds that can be explained by sample identity by fitting a linear model with indicator variables on the sample identity as predictor variables. We used the LinearRegression function of sklearn.linear_model to fit the model, and the 'score' method to evaluate $R^2$, the fraction of the variance explained by this model.

Turn angle distributions for each speed class were computed by collecting all relative angles between consecutive segments amongst cells in that speed class; the distributions were symmetric about θ = 0 and so were folded to be between 0 and π radians. 95% confidence intervals were calculated based on a bootstrap over trajectories in each speed class. For the relationship between local speed and turn angles (*Figure 2B*), the local speed was estimated as the average speed of the two consecutive steps surrounding a turn. Turns were binned based on the local speed, and the average of the cosine of the turn angles was computed for each bin. For this and other binned statistics, the x location of the bin was fixed to be the average value for the points in that bin.

To estimate the rate of speed switching, all trajectories of at least 117 min in length were used. The average speed of each trajectory was measured on 19.5 min intervals; 19.5 min was chosen to minimize the bias-variance trade-off. Specifically, because every cell samples speeds from a distribution, there is trade-off between measuring speeds on intervals that are too short, which may not

give a good estimate of the mean, and intervals that are too long, where cells may switch during the interval. To minimize this trade-off, the interval that maximized the rank correlation between adjacent non-overlapping blocks was used. The average speed of each cell was measured on non-overlapping intervals, and the Spearman rank correlation coefficient between all pairs of intervals was computed. The correlation as a function of time was calculated as the average over all pairs of intervals with the same difference in start times. We computed 95% confidence on a bootstrap over trajectories. For the null model, we permuted speeds amongst the trajectories on each interval; we calculated 95% confidence intervals over the permutations.

For *Figure 3*, the average of the cosine of turn angles between adjacent steps was calculated for each trajectory, as well as the average over all adjacent steps of the secant approximated speeds. For *Figure 3D*, the correlation between local speed and turns was computed as the Pearson correlation coefficient between the local speed, as defined above, and turn angles across the set of adjacent steps in the trajectory.

To estimate the fraction of the variance in turning behavior explained by the cell speed, we fit a spline curve (UnivariateSpline class of scipy 1.3.0; default parameters) to the relationship between speed and the average of the cosine of the turn angles (*Figure 3—figure supplement 1A*). Letting the spline function be $f$, we estimated the variance accounted for by the speed as $V_s = Var(f(S_m))$, where the index $m$ labels trajectories. We estimated the variance in the means due to variation within trajectories, which we called stochasticity, as $V_{st} = \frac{1}{n}\sum_{m=1}^{n}\frac{1}{k_m-1}Var_j(\cos\theta_{jm})$, where $n$ is the total number of trajectories, $k_m$ is the number of turn angles within trajectory $m$, and $\cos\theta_{jm}$ is the cosine of the turn angle $j$ in trajectory $m$. Remaining variance we classified as other (*Figure 3—figure supplement 1B*); this may be due to imperfections in the spline model, other experimental noise, or additional biological variability.

The persistence time was defined to be the time elapsed before the trajectory turns at least $\frac{\pi}{2}$ radians, averaged along the trajectory. Specifically, letting the displacement between timepoints $s$ and $s+1$ be $\vec{x}(s)$, the persistence time along each trajectory was calculated as:

$$\tilde{P} = \frac{1}{n}\sum_{s=1}^{n}\tau(s), \ \ \tau(s) = \sum_{t=s}^{m-1} t_{int} \tag{10}$$

where $m>s$ is the first timestep for which $\vec{x}(s)\cdot\vec{x}(m)<0$, $t_{int}$ is the time interval, and $n$ is the final base point for which $m \leq N$, where $N$ is the final timepoint. For *Figure 4A*, trajectories were binned into mean speed deciles, and the average persistence time was calculated over trajectories in the bin; error bars represent 95% confidence intervals on a bootstrap over trajectories. We also repeated this analysis to measure the average time elapsed before the trajectory turns at least $\frac{\pi}{6}$ radians (*Figure 4—figure supplement 2*).

The effective diffusion coefficient at time $\tau$ was measured as:

$$D_{eff}(\tau) = \frac{MSD(\tau)}{4\tau}. \tag{11}$$

To measure $D_{eff}(\tau)$ as a function of $S$, cells were divided into speed bins (with 5% of the speed distribution per bin); $D_{eff}(\tau)$ for each speed bin was measured by averaging the $D_{eff}(\tau)$ across trajectories, and error bars were computed based on a bootstrap over all trajectories. Note that $D_{eff}(\tau)$ will be independent of $\tau$ only if diffusion scaling is respected, so that the collapse of the data in *Figure 4B* is additional corroboration that the trajectories behave diffusively at long times.

As shown in the models section of the SI below, under the UPT model:

$$D_{eff} \propto S^2, \tag{12}$$

whereas under the SPC model,

$$D_{eff} \propto S^2(\frac{S}{\alpha}+\beta), \tag{13}$$

where $\alpha$ and $\beta$ are fixed across all trajectories. We fixed $\alpha$ and $\beta$ by fitting a line to the persistence time relationship in *Figure 1C*; note that this is a short-time statistic and need not a priori predict the effective diffusion constant at longer times. We fit the UPT model (dashed line) and the SPC

model (solid) to the measured $D_{eff}$ as a function of $S$; in both cases, there was one fitting parameter which was the constant of proportionality, which allows for an offset on the y-axis in log space but does not change the shape of the curve.

## Analysis of scRNAseq data

Reads were aligned to the Zebrafish reference genome (genome release: GRCz10; annotations: GRCz10.85) using STAR (2.5) *Dobin et al., 2013*; reads aligned to each gene were counted using the htseq-count function of HTseq (0.8.0) *Anders et al., 2015*, with the options -m intersection-non-empty and –nonunique all. Note that the final option counts reads that align to a location with more than one annotated feature (e.g. overlapping ORFs) as belonging to both features. This is necessary because of mis-annotation of the T cell receptor light chain constant region in the zebrafish reference genome; both ENSDARG00000075807 (*traj39*) and ENSDARG00000104132 (*traj28*) contain the *trac*, so that reads mapping to *trac* would otherwise be discarded.

Cells were filtered if they expressed fewer than 650 genes or more than 3250 genes, and if more than 8% of reads were of mitochondrial origin. We used UMAP (0.3.1) with the default options to embed the log-transformed counts table in two dimensions, including all genes expressed in at least 10% of cells; and HDBSCAN (0.8.22) with min_samples = 10 to call clusters (*Figure 6—figure supplement 1A*). Comparison with the index sort data (*Figure 6—figure supplement 1B*) showed that cells from the larger cluster had FSC-BSC consistent with lymphocytes, whereas cells from the other cluster had higher FSC and BSC. We called the major cluster as the first cell group, excluding the 4 cells with $\mathrm{BSC} > 2 \times 10^5$, and other cells as the second group. We measured differential expression of genes between the two clusters as: $D = \frac{1}{n} \sum_{i=1}^{n} \log_2(E_{1i}) - \frac{1}{m} \sum_{i=1}^{m} \log_2(E_{2i})$, where $E_{1i}$ are expression values, in counts per million (CPM) + 1, for the first cluster, $n$ is the number of cells in this cluster, $E_{2i}$ are expression values (in CPM+1) for the second cluster, and $m$ is the number of cells in this cluster. (Note that this is the log of the ratio of the geometric means within each cluster.) In *Supplementary file 2*, we report the genes with $D > \log_2 10$ and $p < .01$ (Wilcoxon rank-sum test, Bonferonni-corrected.) Genes enriched in the larger cluster included the T cell and immune-related genes *tagapa*, *tagapb*, *ccr9a*, *tnfrs9b*, *il2rb*, and *ptprc* (*Tabula Muris Consortium et al., 2018*); we also tested for expression of the T cell receptor light chain constant region (*trac*; expression estimated based on the expression of ENSDARG00000075807). Based on these markers, we identified the larger group (n = 330) as T cells. The significantly differentially expressed genes enriched in this group also included *arpc1b*, *wasb*, *arhgdig*, *coro1a*, *scinlb*, and *capgb*, which we classified as belonging to the WASP/ARP2/3 pathway based on the literature (*Vicente-Manzanares et al., 2002*). Finally, we observed very little expression of markers associated with other types of immune cells (the B cell light chain *igic1s*, the B cell marker *ccl35.2*, and the neutrophil and macrophage markers *mpeg1* and *mpx* (*Tang et al., 2017*) in either group (*Figure 6B*, *Figure 6—figure supplement 2*). The genes most significantly enriched in the non-T cell group include keratin proteins (*krt8*, *KRT1*), as well as *ahnak* (*Figure 6—figure supplement 2*, *Supplementary file 2*). Based on these markers, we identified these cells as epithelial cells, possibly keratinocytes (*Tabula Muris Consortium et al., 2018*). We note that we observed GFP signal in the somite region of the Tg(*lck*:GFP) tail via microscopy (see, e.g., *Figure 1A* and Movie S1) which we did not observe in wildtype *nacre*[-/-] zebrafish, suggesting that these cells may mis-express the marker.

To analyze finer-scale variation within the T cell cluster, we used the self-assembling manifold algorithm method from *Tarashansky et al., 2019*. We first used the SAM algorithm with parameters thresh = 0.1 and k = 10 to perform dimensional reduction. Visualization of a UMAP projection together with the labels from our sort plates (*Figure 6—figure supplement 3*) showed that there was a plate effect related to p1; we excluded this plate from the remainder of the analysis. We re-ran the SAM algorithm using the remaining (n = 237) T cells, with parameters thresh = 0.1, k = 10, to produce the dimensional reduction shown in *Figure 6A*. Cluster labels were assigned using the KMeans class in sklearn.cluster. We measured differential expression of genes between the two subtypes as described above. In *Supplementary file 1*, we report the genes with $D > \log_2 10$ and $p < .01$ (Wilcoxon rank-sum test, Bonferonni-corrected), for the two subtypes. In *Figure 6B*, we show a subset of markers associated with the whole T cell cluster, as well as selected marker genes between the two subtypes.

To identify potential co-variation with *arpc1b* expression amongst the T cells, we computed the Spearman rank correlation between expression of *arpc1b* (in counts per million) and all other genes expressed in at least 20% of cells. In each calculation of pairwise correlation coefficient, we excluded cells with zero counts for both genes to avoid biasing correlations upwards due to spurious points at the origin. We calculated p values using a permutation test: we permuted cell labels and re-calculated all correlation coefficients; p-values were calculated as the proportion of observations of a correlation coefficient higher than the observed coefficient, multiplied by the number of genes tested (Bonferonni correction).

## Acknowledgements

We acknowledge Aya Ludin-Tal and Leonard Zon for the generous gift of the Tg(*lck*:GFP) zebrafish line. We acknowledge Kiran Kocherlakota and the Stanford VSC for assistance with zebrafish management and husbandry. We acknowledge Saroja Korullu for assistance with library preparation. We also acknowledge Stanford Research Computing and the Sherlock2 computer cluster for computational support and resources. Finally, we acknowledge Louis Leung and Karen Mruk for invaluable advice regarding microscopy and zebrafish; Edward Marti for helpful discussions on imaging, analysis, and the manuscript, and Felix Hornes and Michael Swift for comments on the manuscript. Funding: This project was supported by the Chan Zuckerberg Biohub Competing interests: Authors declare no competing interests. Data and materials availability: Sequencing data and the gene expression count table have been deposited on GEO (accession: GSE137770). Analysis code and trajectory data are available at https://github.com/erjerison/TCellMigration.

## Additional information

### Funding

| Funder | Author |
| --- | --- |
| Chan Zuckerberg Biohub | Elizabeth R Jerison<br>Stephen R Quake |

The funders had no role in study design, data collection and interpretation, or the decision to submit the work for publication.

### Author contributions

Elizabeth R Jerison, Conceptualization, Resources, Software, Formal analysis, Investigation, Visualization, Methodology, Writing - original draft; Stephen R Quake, Conceptualization, Supervision, Funding acquisition, Writing - original draft

### Author ORCIDs

Elizabeth R Jerison ⓘ https://orcid.org/0000-0003-3793-8839
Stephen R Quake ⓘ https://orcid.org/0000-0002-1613-0809

### Ethics

Animal experimentation: This study was performed in strict accordance with the recommendations in the Guide for the Care and Use of Laboratory Animals of the National Institutes of Health. All of the animals were handled according to approved institutional animal care and use committee (IACUC) protocols (#32107) of Stanford University.

### Decision letter and Author response

Decision letter https://doi.org/10.7554/eLife.53933.sa1
Author response https://doi.org/10.7554/eLife.53933.sa2

# Additional files

## Supplementary files

• Supplementary file 1. Differentially-expressed genes between the two T cell sub-clusters (*Figure 6A*). Log differential expression ratio (see Materials and methods) and Bonferroni-corrected Wilcoxon rank-sum p-value are listed for each gene. Genes with at least 10-fold differential expression and Bonferroni-corrected Wilcoxon rank-sum p-value <.01 are included.

• Supplementary file 2. Differentially-expressed genes between the T cells and putative epithelial cell clusters (*Figure 6—figure supplement 1*, *Figure 6—figure supplement 2*). Log differential expression ratio (see Materials and methods) and Bonferroni-corrected Wilcoxon rank-sum p-value are listed for each gene. Genes with at least 10-fold differential expression and Bonferroni-corrected Wilcoxon rank-sum p-value <.01 are included.

• Transparent reporting form

## Data availability

Sequencing data have been deposited in GEO under accession code GSE137770. All source data, including cell trajectories, and analysis code are available at: https://github.com/erjerison/TCellMigration (copy archived at https://github.com/elifesciences-publications/TCellMigration).

The following dataset was generated:

| Author(s) | Year | Dataset title | Dataset URL | Database and Identifier |
|---|---|---|---|---|
| Jerison ER, Quake SR | 2019 | Characterization of T cells from the larval zebrafish tail via single-cell RNAseq | https://www.ncbi.nlm.nih.gov/geo/query/acc.cgi?acc=GSE137770 | NCBI Gene Expression Omnibus, GSE137770 |

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

## Appendix 1

# Persistent random walks: the uniform persistence time (UPT) and the speed-persistence coupling (SPC) models

The infinitesimal model most commonly used to describe metazoan cell migration is the Orenstein-Uhlenbeck model (OU) (**Uhlenbeck and Ornstein, 1930**), which has also been called the persistent random walk model (PRW) in the context of cell migration (**Wu et al., 2014**). While we have chosen this model for concreteness, the statistical features discussed below are also in common to a number of other models that include some directional persistence but are diffusive at long times, including the Kratky-Porod wormlike chain model (**Doi and Edwards, 1988**). We briefly review some of the standard results which we use to compare to data below.

Under the OU model, the dynamics of a cell are described by:

$$\frac{dv_i(t)}{dt} = -\frac{1}{P}v_i(t)dt + \frac{S}{\sqrt{P}}\eta_t, \tag{14}$$

where S is the speed parameter; P is the persistence time parameter; $\eta_t$ is a Gaussian white noise term; and $i = x, y, z$. This model, considered the prototypical noisy relaxation process, produces two main qualitative features: trajectories that turn smoothly (i.e. directional persistence), and diffusive behavior at times $t \gg P$. We note that fluctuations in velocity and speed along the trajectory are also features of this model. In particular, the velocity-velocity autocorrelation function is given by:

$$\langle v_i(t)v_i(s)\rangle = S^2 e^{-\frac{|t-s|}{P}}. \tag{15}$$

Setting $t = s$, we see that the speed parameter $S$ is proportional to the root-mean squared speed: $S = \sqrt{\langle \frac{1}{n}\sum_i^n v_i(t)^2\rangle}$, where $n$ is the number of dimensions. Because the distribution of velocities generated by the model is Gaussian, $S$ is also proportional to the mean speed. The decay of the velocity autocorrelation in each component sets the turning timescale at $P$; at long times the directions of motion are uncorrelated.

The mean-squared displacement (MSD) after a time interval $\tau$ is given by:

$$\langle(\vec{x}(\tau) - \vec{x}(0))^2\rangle = 2nS^2 P\tau(1 + \frac{P}{\tau}(e^{-\frac{\tau}{P}} - 1)), \tag{16}$$

where n is the number of dimensions. The MSD scales advectively, as $nS^2\tau^2$, in the limit of $\tau \ll P$, and diffusively, as $2nS^2 P\tau$, in the limit of $\tau \gg P$. Thus the model predicts that $\frac{\langle(\vec{x}(\tau)-\vec{x}(0))^2\rangle}{\tau}$ will approach a constant value of $2nS^2 P$ at long times, which defines the effective diffusion constant to be $\frac{1}{2}S^2 P$.

We refer to the OU model with fixed persistence time parameter $P$ (but potentially variable speed parameters $S$) as the uniform persistence time (UPT) model. We note that under the OU model, the MSD and PSD depend on the speed parameter $S$ only through the constant scale factor $S^2$: for fixed $P$, the quantities $\frac{\langle(\vec{x}(\tau)-\vec{x}(0))^2\rangle}{S^2}$ and $\frac{\langle v_i(f)^2\rangle}{S^2}$ are independent of speed, as are the normalized quantities $\frac{\langle(\vec{x}(\tau)-\vec{x}(0))^2\rangle}{\langle(\vec{x}(\tau_0)-\vec{x}(0))^2\rangle}$ and $\frac{\langle v_i(f)^2\rangle}{\langle v_i(f_0)^2\rangle}$, where $\tau_0$ and $f_0$ are a chosen time interval and frequency, respectively. (Note that this is also true of the full dynamics: we can eliminate the dependence on $S$ by transforming to the variable $\tilde{v} = \frac{v}{S}$, or measuring distance in units proportional to $S$.) In particular, the effective diffusion constant $D_{eff} = \frac{1}{2}S^2 P$ scales with $S^2$.

Our observation of a linear relationship between measured mean speeds and correlation times suggests the following constrained form of the OU model, which we have called the speed-persistence coupling (SPC) model:

$$\frac{dv_i(t)}{dt} = -\frac{1}{\frac{S}{\alpha}+\beta}v_i(t) + \frac{S}{\sqrt{\frac{S}{\alpha}+\beta}}\eta_t,$$ (17)

where $\alpha$ is a constant with units of acceleration; $\beta$ is a constant with units of time; and both are fixed across all cells.

In this model, the effective diffusion constant is:

$$D_{eff} = \frac{1}{2}S^2(\frac{S}{\alpha}+\beta).$$ (18)

Under the SPC model, the control parameter $S$ is proportional to the cell's mean speed, so that this observable fully specifies its dynamics.

Finally, we note that with fixed $S$ and $P$, these models still produce variation in both local speed and turning behavior along trajectories; the control parameters, together with *Equation 14*, set the distributions of these quantities.

## Effects of finite-length trajectories, sampling intervals, noise, and distributions of persistence time on speed-persistence coupling

Measurements of speed and persistence time are imperfect estimators of the underlying continuous process. Here we address whether statistical artifacts could generate the observed correlations between speed and persistence time. In particular, the finite sampling interval introduces a bias downwards in all speed estimates, because some turns are missed. Because this effect is stronger for less-persistent cells, which turn more, we expect it to introduce a correlation between the measured speed and the measured persistence time.

To evaluate the influence that this may have had on our data, we simulated a collection of cells with the same speeds as our measured cells under the OU model. Simulations were performed using the velocity update rule in *Equation 14* (*Gillespie, 1996*), with 20 simulated intervals $dt$ per sampling interval. Position coordinates were determined by numerical integration of the velocities along the simulated trajectories. Noise in centroid locations was included by adding a Gaussian random variable to x and y positions. We conservatively set the noise parameter at $\sigma = 3\,\mu m$ per sampling interval; this was chosen as an estimate of the combined effects of true technical noise and changes in cell shape during the interval. Each simulated trajectory was 50 sampling intervals (1000 microscopic timesteps) in length. To match the measurement, sampling intervals were assigned to be 45 s in length. We measured both mean speeds and correlation times on the simulated trajectories as defined in Materials and methods.

We first assessed whether the SPC model, defined in the previous section, with the addition of noise in the centroid locations, gave the expected dependence of measured persistence time on measured cell speed (*Appendix 1—figure 1B*). Next, we simulated a collection of cells with the same set of speeds as in the data, with a constant persistence time parameter $P$ (the UPT model), to check whether the finite length of the trajectories induced a correlation between measured speeds and persistence times (*Appendix 1—figure 1C*). We did not observe a significant effect. Next, we added centroid location noise to the UPT model. The addition of noise does induce a correlation at the slow end of the speed spectrum (*Appendix 1—figure 1D*); this is because cells that happen to have turned more will appear slower. However, this effect becomes negligible for cells with speeds above the noise level. We note that this likely contributes to the measured propensity for sharp turns amongst the slowest cells, and may lead to misassignment between the two lowest speed classes.

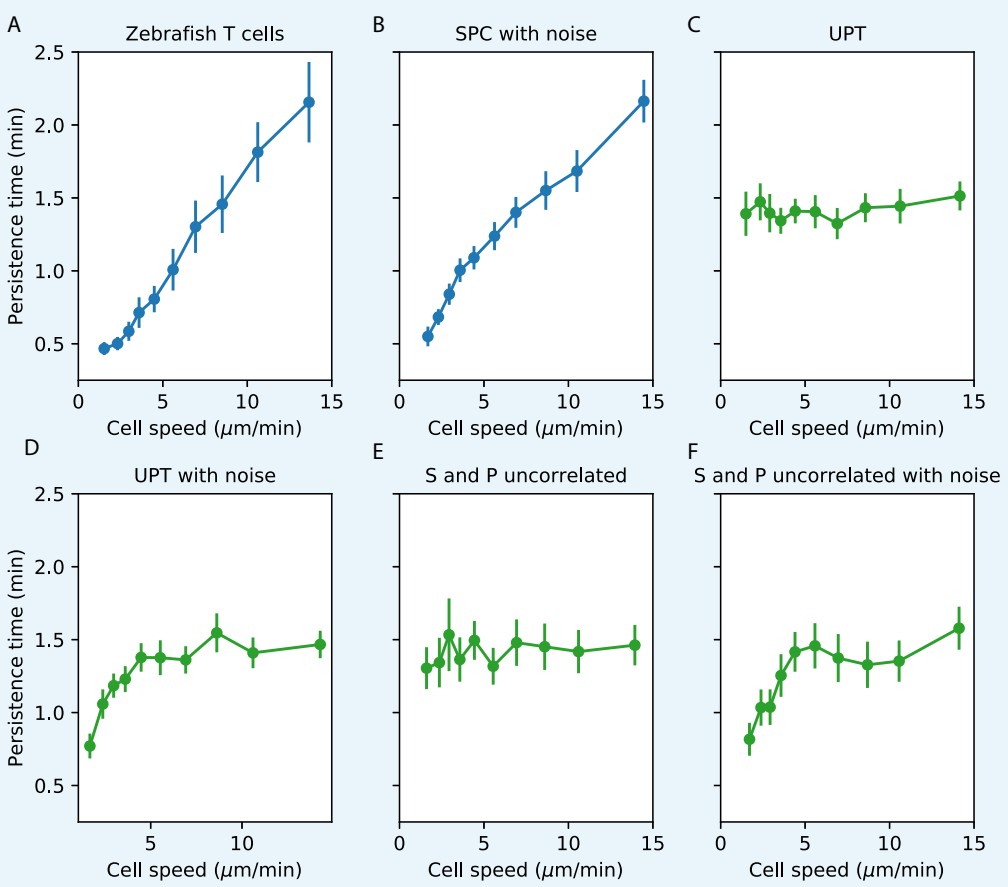

**Appendix 1—figure 1.** Comparisons between speed-persistence time relationship in simulations and data. (**A**) Data (as in *Figure 4A*). (**B**) Simulation of the SPC model with empirical parameters (see Appendix 1 for details). (**C**) Simulation of uniform persistence time (UPT) model, with speeds and persistence times measured as in the data. Biases introduced by measured speeds and persistence times do not lead to an observable correlation. (**D**) Simulation of UPT model with a conservative estimate of mislocation noise. A spurious correllation is induced at low cell speeds, but cannot account for the trend across cell speeds that we observe. (**E**) Simulation of a model where the predicted persistence times have been reshuffled amongst the cells, to simulate an empirically-realistic model where persistence times vary but are uncorrelated with speed. This does not generate a significant bias in the speed-persistence relationship. (**F**) Model with reshuffled persistence times, as in E., and mislocation noise. As in D., this leads to a spurious correlation at low speeds but no other significant effects.

We next evaluated whether a model where both $S$ and $P$ varied in a manner consistent with the data, but were uncorrelated with each other, could induce a correlation between measured speed and persistence time. Such a correlation could appear on the faster end of the speed spectrum due to variable $P$, because the fastest measured cells are biased to having been both fast and particularly persistent. We evaluated the size of this effect in our data by simulating a collection of cells with the observed speed distribution, as before, permuting the predicted $P$ parameters from the SPC model amongst the simulated cells. We found that this did not measurably bias the speed-persistence relationship (*Appendix 1—figure 1E*). Finally, we simulated the uncorrelated $S$ and $P$ model with the addition of centroid location noise (*Appendix 1—figure 1F*).

From this analysis, we concluded that noise in the locations creates a spurious correlation between measured speed and measured persistence time at the slow end of the speed spectrum, but that this effect cannot account for the consistent correlation across speeds that

we observe; and that a model with variable $P$ that is uncorrelated with $S$ also cannot account for our observations.

## Potential effects of fin fold boundary on MSD

As noted in the text, we observed that trajectories tended to stay within the tail fin and larval fin fold (*Figure 1—figure supplement 1*), without crossing into the interior muscle region, suggesting a partial barrier between these zones. In principle, such a barrier could lead to saturation of the MSD. If present, this effect would be strongest for trajectories in the regions of the fin fold on the margins of the fish (*Figure 1—figure supplement 1*). To assess whether this effect could be driving the observed transition away from advective/superdiffusive behavior, we took two approaches. First, we note that the typical timescale between turns (*Figure 4A*) is 2 min, and diffusive scaling is reached by $\tau = 9$ min (*Figure 1C* inset, *Figure 4B*). Over 9 min, the median cell experiences an average displacement of 15 $\mu m$, while the 80th percentile cell experiences an average displacement of 40 $\mu m$. Given that the width of the fin fold is about 200 $\mu m$ (although sometimes narrower, *Figure 1—figure supplement 1*), we would not expect boundary effects to be significant at the transition timescale. To further assess this, we calculated the MSD as a function of time lag on the control sample shown in *Figure 2A–B* and *Figure 2—video 1*. This sample includes only the tail fin region, without fin fold boundary effects. Data for this sample was acquired at a slightly different time interval (see Materials and methods), and so it was not pooled into the overall MSD calculation. We calculated MSD as a function of time lag for this sample and compared it to the MSD calculated using all other control samples, and found very good agreement (*Appendix 1—figure 2*). (Note that the error bars are 95% confidence intervals calculated on a bootstrap over trajectories; the larger error bars for the held-out sample reflect the smaller number of trajectories.) We thus concluded that barrier effects on the margins of the fin fold did not drive the transition we observed in the MSD.

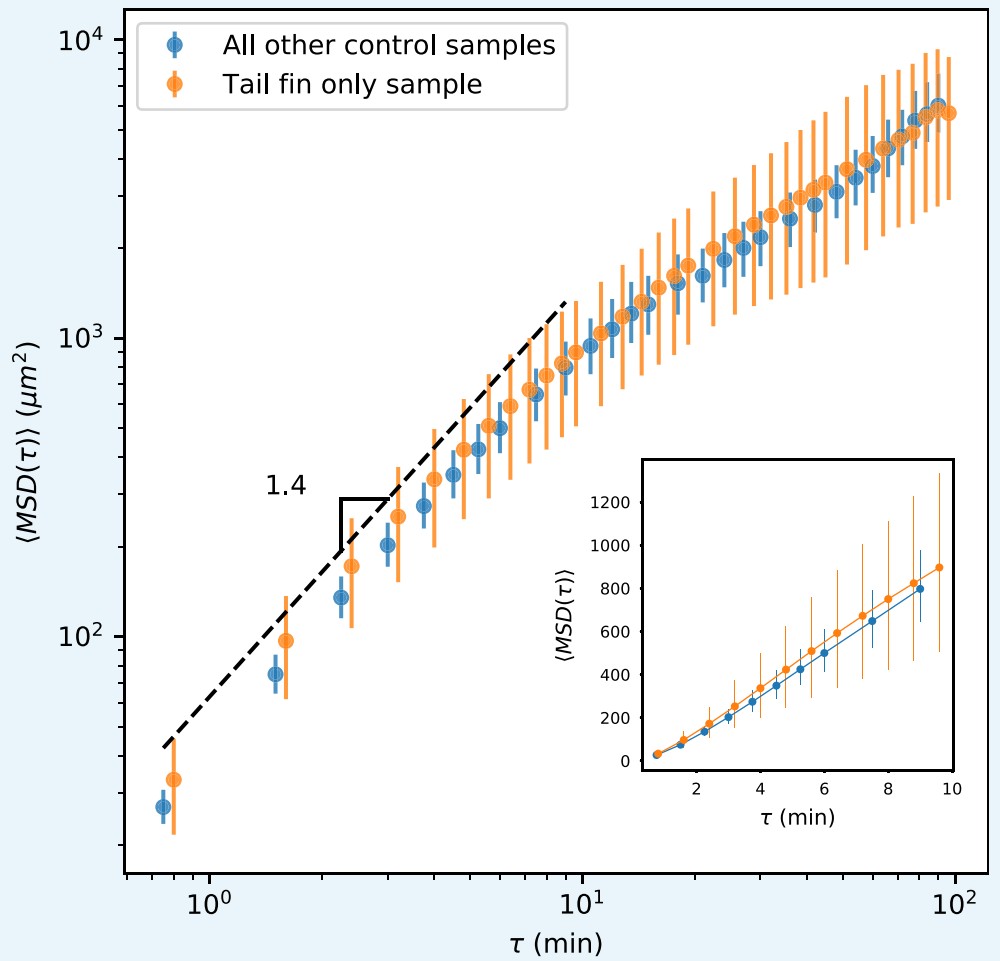

**Appendix 1—figure 2.** MSD measured on all control trajectories except those from the sample shown in *Figure 2*, compared with MSD measured on the trajectories from the held-out sample. All trajectories for this sample were measured in the tail fin area, and so are not subject to potential effects due to a boundary between the fin fold and muscle region of the tail (*Figure 1—figure supplement 1*). We do not observe a difference between these measurements. Error bars represent 95% confidence intervals on a bootstrap over trajectories.

