## [Decision Letter]

**Acceptance summary:**

The paper by Jerison and Quake characterizes T-cell motility in tissues that span a broad range of length scales from microns, when contacting APCs, to millimeters, relevant for finding rare antigens. With sophisticated experiments the authors tracked populations of T-cells in live larval fish over several hours and built a theoretical model that relates persistence and speed of T-cells. This work is a significant contribution that will enable better understanding of immune responses in vivo.

**Decision letter after peer review:**

Thank you for submitting your article "Heterogeneous T cell motility behaviors emerge from a coupling between speed and turning in vivo" for consideration by *eLife*. Your article has been reviewed by three peer reviewers, and the evaluation has been overseen by a Reviewing Editor and Aleksandra Walczak as the Senior Editor. The following individuals involved in review of your submission have agreed to reveal their identity: Judy Cannon (Reviewer #2); Michael L Dustin (Reviewer #3).

The reviewers have discussed the reviews with one another and the Reviewing Editor has drafted this decision to help you prepare a revised submission.

Summary:

The manuscript characterizes T-cell motility in tissues. T-cells explore tissues over a wide range of scales that span from microns (e.g., when contacting APCs) to millimeters (e.g. relevant for finding rare antigens). An important point is made that a simple diffusion process is not sufficient to explain this broad range of behaviors in T-cell motility. Two possible solutions are discussed in the manuscript: Anomalous diffusion via cells making Levy walks, or regular random walks sampled from a wide distribution. Extensive and sophisticated experiments are conducted on the native population of T cells in live larval fish over several hours, imaging millimeter fields of view. By analyzing many cell trajectories, the authors show that, at least in the stationary state, the motility of T-cells seem to be consistent with a regular random walk sampled from a wide distribution. Interestingly, the data spans over an effective 1D manifold in the space of T-cell speed and persistence. Moreover, the authors observe similar statistics for cell motility in mouse and in *Dictyostelium* (from previously published data), pointing towards a more generic pattern of motility that could be thought as a population strategy to span a broad range of scales -- one is tempted to call this "motility bet-hedging".

Overall, the reviewers agree that the experiments present a significant improvement to the existing methods in recording cell motility and that the manuscript provides a sufficient theoretical novelty in characterizing T-cell motility strategies. However, there are some concerns with the presentation and the analyses that we would like to see addressed.

Essential revisions:

1) There is a possibility that the cell-to-cell variation can be associated with distinct behaviors of different T-cell subtypes. We suggest that the authors use single cell gene expression levels (reported in Figure 5) to investigate whether the differential expression of certain surface markers is predictive of the mode of cell motility. Perhaps there can even be some memory cells in the pool. This would not require more experiments but some more analyses, which should be doable within a reasonable time frame. In principle, this analysis could strongly add to the mechanistic understanding of the reported bet-hedging strategy in cell motility.

2) For analysis of persistence, the authors state that pi/2 radians before turning was used as a definition for a persistent cell. This may be too broad as cells that turn 60o would likely not be classified as persistent. The authors should re-analyze persistence using 15o or 30o as a cutoff to ensure that motility analysis holds with a more stringent definition of persistent motility.

3) The experimental detail as written in the current version is insufficient. For example, are T cells vascular or in tissues? While the authors state that "tail fin" is imaged, there is no description of how specific areas are selected to image. Why was the tail fin selected? Was there an infection or was all motility in the absence of infection? Also, T cells at 10-12 dpf were imaged. This is an extremely early time point. The authors should explain why this time point is used rather than imaging T cell motility in adult fish (transparent zebrafish for imaging are available).

4) Previous work by Mairui and colleagues (Cell 2015) has shown the coupling between cell speed and persistence across many cell types. The authors seem to be unaware of this study.

5) Authors should discuss some of the other models for persistent random walks that have been recently considered in the field of active matter. Given the Maiuri et al's finding (Cell:2015) that actin flow and actin regulatory proteins drive cell motility, the non-equilibrium models for random walk could provide a better description for this process.

6) The authors reject that cell motility is a levy walk, which was previously proposed by Harris et al., 2012, for CD8^+^ Tcells. The conditions of the two experiments are not comparable as as Harris et al. studied CD8^+^ cells chasing antigens, whereas the analyses in this manuscript are based on cell trajectories in the stationary state. This should more clearly be pointed out.

7) Authors should include a discussion on prior related work:

i) Negulescu PA, et al. Polarity of T cell shape, motility, and sensitivity to antigen. Immunity. 1996;4(5):421-30.

ii) Dong TX, et al. Intermittent Ca(2+) signals mediated by Orai1 regulate basal T cell motility. *eLife*. 2017;6. doi: 10.7554/*eLife*.27827.

iii) Harris TH, et al. Generalized Levy walks and the role of chemokines in migration of effector CD8^+^ T cells. Nature. 2012;486(7404):545-8.

iv) Maiuri P , et al. Actin flows mediate a universal coupling between cell speed and cell persistence. Cell 2015:161(2):374-86

8) A more detailed discussion is necessary on how the proposed search strategy in the manuscript compare with Levy or intermittent migration (Maiuri et al., 2015; Harris et al., 2012).

---

## [Author Response]

Essential revisions:1) There is a possibility that the cell-to-cell variation can be associated with distinct behaviors of different T-cell subtypes. We suggest that the authors use single cell gene expression levels (reported in Figure 5) to investigate whether the differential expression of certain surface markers is predictive of the mode of cell motility. Perhaps there can even be some memory cells in the pool. This would not require more experiments but some more analyses, which should be doable within a reasonable time frame. In principle, this analysis could strongly add to the mechanistic understanding of the reported bet-hedging strategy in cell motility.

This is a great suggestion. While it is not technologically feasible at this time to associate an individual cell (and its trajectory) with a single-cell RNA expression profile, we can nonetheless use the single-cell RNA sequencing data to ask whether there is covariation in motility genes amongst the T cells. We performed additional analysis of the single-cell RNA sequencing data, and found that there are two main subgroups of cells (Figure 6A-B). Additionally, we found statistically significant covariation of actin remodeling genes from cell to cell (Figure 6C-D). Interestingly, although distinct in other dimensions of gene expression space, the two cell subtypes have overlapping, continuous distributions of the actin-remodeling related genes. This analysis suggests that there is real cell-to-cell variation in actin nucleation at the transcriptional level, which may induce long-lived heterogeneous motility states. It is also consistent with a continuous motility axis. We note that speed and other motility characteristics may well also be regulated post-transcriptionally. We have included the results of this analysis as the final subsection of the Results.

Regarding the presence of memory cells: unfortunately the incomplete homology between zebrafish and mammalian systems makes it difficult to draw firm conclusions regarding T cell subtypes. For example, memory T cells are often classified with respect to their expression of CD45 isotypes. While the zebrafish homologue of CD45 (ptprc) is highly expressed, the exon structure of the zebrafish gene is not readily mappable to its mammalian counterpart. Additionally, even for the much better studied mammalian systems, there is currently significant controversy regarding the relationship between the traditional cell-surface markers and single-cell transcriptional profiles and how to use the latter to classify T cell subtypes. We expect that the rapidly-increasing volume and discussions around scRNAseq from immune cells will clarify this significantly over the next few years, but for the present we believe there is not enough information to confidently annotate T cell subtypes.

2) For analysis of persistence, the authors state that pi/2 radians before turning was used as a definition for a persistent cell. This may be too broad as cells that turn 60o would likely not be classified as persistent. The authors should re-analyze persistence using 15o or 30o as a cutoff to ensure that motility analysis holds with a more stringent definition of persistent motility.

Thanks for the suggestion. We have repeated the analysis of persistence times as a function of cell speed with 30 degrees as a cut-off for persistence (Figure 4—figure supplement 2). We find a similar linear relationship between speed and persistence, with, as expected, shorter times over which cells are persistent under this definition.

3) The experimental detail as written in the current version is insufficient. For example, are T cells vascular or in tissues? While the authors state that "tail fin" is imaged, there is no description of how specific areas are selected to image. Why was the tail fin selected? Was there an infection or was all motility in the absence of infection? Also, T cells at 10-12 dpf were imaged. This is an extremely early time point. The authors should explain why this time point is used rather than imaging T cell motility in adult fish (transparent zebrafish for imaging are available).

Thanks for the comments. We have clarified these points in the manuscript. The T cells are in tissue; there are also T cells in circulation, but they move orders of magnitude faster and are therefore not captured in our analysis. (Occasionally one is caught in a frame; see e.g. Figure 2—video 1, at 0:19:00 on the timestamp in the upper left.) All motility was measured in the absence of infection. We imaged as much as possible of the region of the fish posterior to the anus; this region of the fish is composed primarily of the tail fin and larval fin fold (annotated in Figure 1—figure supplement 1). We chose this region because it enabled us to image cell behavior within a millimeter-scale tissue, while also avoiding the highly autofluorescent gut, which overwhelmed signal from the cells. We note that we have also added a short section (subsection “Model predicts wide variation in length scales of exploration across the population” penultimate paragraph and Appendix 1 final subsection and figure) addressing any potential effects of the fin fold boundary. Finally, it is currently infeasible to image older fish for long timespans because once the gills fully develop, the fish must pump their gills to remain oxygenated, which is incompatible with immobilization for light sheet microscopy. (At earlier timepoints, the fish acquire oxygen through passive diffusion via the skin.) Some intubation platforms for adult zebrafish have been developed, but no such technology exists yet for light sheet microscopy.

4) Previous work by Mairui and colleagues (Cell 2015) has shown the coupling between cell speed and persistence across many cell types. The authors seem to be unaware of this study.

Thanks very much for pointing out this highly relevant paper! We were indeed unaware. We have modified our main text and discussion to include this paper.

5) Authors should discuss some of the other models for persistent random walks that have been recently considered in the field of active matter. Given the Maiuri et al's finding (Cell:2015) that actin flow and actin regulatory proteins drive cell motility, the non-equilibrium models for random walk could provide a better description for this process.

Related to this point and comments 7 and 8 below, we have added discussion of the role of intermittency and alternative models of the random walk process to the Discussion.

6) The authors reject that cell motility is a levy walk, which was previously proposed by Harris et al., 2012, for CD8^+^ Tcells. The conditions of the two experiments are not comparable as as Harris et al. studied CD8^+^ cells chasing antigens, whereas the analyses in this manuscript are based on cell trajectories in the stationary state. This should more clearly be pointed out.

We have added this clarification to the manuscript. (Discussion section).

7) Authors should include a discussion on prior related work:i) Negulescu PA, et al. Polarity of T cell shape, motility, and sensitivity to antigen. Immunity. 1996;4(5):421-30.ii) Dong TX, et al. Intermittent Ca(2+) signals mediated by Orai1 regulate basal T cell motility. eLife. 2017;6. doi: 10.7554/eLife.27827.iii) Harris TH, et al. Generalized Levy walks and the role of chemokines in migration of effector CD8^+^ T cells. Nature. 2012;486(7404):545-8.iv) Maiuri P , et al. Actin flows mediate a universal coupling between cell speed and cell persistence. Cell 2015:161(2):374-86

Thanks for the references. In response to this comment as well as points 3 and 7, we have added to the Discussion to comment on how intermittency manifests in our model and its relationship with prior work.

8) A more detailed discussion is necessary on how the proposed search strategy in the manuscript compare with Levy or intermittent migration (Maiuri et al., 2015; Harris et al., 2012).

Thanks for the suggestion. The question of optimality of different search strategies depends sensitively on the distribution (sometimes called patchiness) of targets, which is generally unknown in in vivo contexts. Exploring this question in depth would thus require extensive simulations of different possible target distributions, which is beyond the scope of our present study. However, we have added comments on this to the manuscript. (Discussion section).